# The mechanosensitive Piezo1 channel mediates heart mechano-chemo transduction

Fan Jiang[1], Kunlun Yin[2], Kun Wu [1,5], Mingmin Zhang[1], Shiqiang Wang[3], Heping Cheng [4], Zhou Zhou[2] & Bailong Xiao [1✉]

The beating heart possesses the intrinsic ability to adapt cardiac output to changes in mechanical load. The century-old Frank–Starling law and Anrep effect have documented that stretching the heart during diastolic filling increases its contractile force. However, the molecular mechanotransduction mechanism and its impact on cardiac health and disease remain elusive. Here we show that the mechanically activated Piezo1 channel converts mechanical stretch of cardiomyocytes into $Ca^{2+}$ and reactive oxygen species (ROS) signaling, which critically determines the mechanical activity of the heart. Either cardiac-specific knockout or overexpression of Piezo1 in mice results in defective $Ca^{2+}$ and ROS signaling and the development of cardiomyopathy, demonstrating a homeostatic role of Piezo1. Piezo1 is pathologically upregulated in both mouse and human diseased hearts via an autonomic response of cardiomyocytes. Thus, Piezo1 serves as a key cardiac mechanotransducer for initiating mechano-chemo transduction and consequently maintaining normal heart function, and might represent a novel therapeutic target for treating human heart diseases.

[1] State Key Laboratory of Membrane Biology, Tsinghua-Peking Center for Life Sciences, Beijing Advanced Innovation Center for Structural Biology, IDG/McGovern Institute for Brain Research, School of Pharmaceutical Sciences, Tsinghua University, Beijing 100084, China. [2] State Key Laboratory of Cardiovascular Disease, Beijing Key Laboratory for Molecular Diagnostics of Cardiovascular Diseases, Center of Laboratory Medicine, Fuwai Hospital, National Center for Cardiovascular Diseases, Chinese Academy of Medical Sciences and Peking Union Medical College, Beijing 100037, China. [3] State Key Laboratory of Membrane Biology, College of Life Sciences and Institute of Molecular Medicine, Peking University, Beijing, China. [4] State Key Laboratory of Membrane Biology, Institute of Molecular Medicine, Peking-Tsinghua Center for Life Sciences, Peking University, Beijing, China. [5] Present address: Medical Research Center, Beijing Key Laboratory of Cardiopulmonary Cerebral Resuscitation, Department of Emergency, Beijing Chao-Yang Hospital, Capital Medical University, Beijing, China. ✉email: xbailong@mail.tsinghua.edu.cn

Heart experiences drastic mechanical changes on a beat-to-beat basis and evolves intrinsic mechanisms to adapt cardiac output to hemodynamic conditions[1]. Stretching the ventricular wall due to an increase in end-diastolic volume leads to an immediate enhancement of cardiac contraction, followed by a slow response lasting for minutes. These responses in the intact heart have been respectively described as the Frank–Starling law and the Anrep effect for over 100 years[2–4], which constitute powerful mechanisms to allow the heart to adapt to an abrupt rise in either preload or afterload. While the Frank-Starling phenomenon is in part due to a sensitization of the myofibrillar $Ca^{2+}$ sensitivity[5,6], an increase in intracellular $Ca^{2+}$ might play a key role in mediating the stretch-induced biphasic enhancement of cardiac contraction[7,8]. It has been proposed that stretch-activated cation channels might provide a conceptually simple mechanism for conferring cardiac mechanosensitivity and the resulting $Ca^{2+}$ signaling[9,10], which determines the strength of cardiac contraction[11]. However, the molecular identify of the underlying mechanotransduction channel and its role in normal and diseased hearts have remained unclear.

Piezo1 is a bona fide mechanosensitive cation channel[12–15] that mediates mechanically activated cationic currents and $Ca^{2+}$ signaling in various cell types[16–19], including endothelial cells, red blood cells, smooth muscle cells, epithelial cells, osteoblasts, and cardiac fibroblasts[20–26]. Piezo1 is involved in various aspects of vascular physiology[27], including blood and lymphatic vessel development[20,21,28–31], vascular tone[23,32], arterial remodeling[22], and red blood cell volume homeostasis[24]. Furthermore, together with its homolog member Piezo2, Piezo1 has been proposed as the mechanosensor in baroreceptor neurons for baroreflex control of blood pressure and heart rate[33]. However, the expression and role of Piezo1 in cardiomyocytes and heart function have not been genetically studied.

Here, we use mouse genetics and pharmacology to demonstrate the critical role of Piezo1 in cardiac mechano-chemo transduction in normal and diseased hearts.

## Results

**Expression and localization of Piezo1 proteins in cardiomyocytes.** Previous studies have reported relatively low level of *Piezo1* mRNA expression in heart tissues and primarily cultured cardiomyocytes[12,26,34]. However, the abundance and localization of Piezo1 proteins in cardiomyocytes have not been systematically characterized. To enable detection of Piezo1 proteins expressed in mouse hearts, we used the CRISPR/Cas9 technology to generate a Piezo1-Flag-knock-in mouse line (Piezo1-Flag-KI) (Supplementary Fig. 1a, b), in which the Flag-coding sequence was inserted into the genetic sequence coding the mouse *Piezo1* (Supplementary Fig. 1b). Importantly, the Flag-tagged Piezo1 (Piezo1-Flag) proteins were specifically detected in the Piezo1-Flag-KI heart tissue either via western blotting of the anti-Flag-immunoprecipitated sample (Supplementary Fig. 1c) or via immunostaining with the anti-Flag antibody (Supplementary Fig. 1d). As a control, heart tissues derived from the wild-type littermate control mice showed no Piezo1-Flag expression (Supplementary Fig. 1c, d). Interestingly, immunostaining of primarily cultured adult ventricular myocytes using the anti-Flag antibody revealed punctate and striated expression pattern of the Piezo1-Flag protein specifically in the Piezo1-Flag-KI cells, but not in the control cells (Fig. 1a), which was similar to the expression pattern of the Z-line marker protein actinin (Fig. 1a), indicating the localization of Piezo1 near the T-tubule. The endogenous expression of Piezo1 in cardiomyocytes was further confirmed using the previously reported Piezo1-tdTomato knock-in mouse line (Piezo1-tdTomato-KI) (Fig. 1b), in which the tdTomato coding sequence was knocked into the genetic sequence coding the C-terminus of the mouse Piezo1 protein[20]. Immunostaining using the anti-dsRed antibody clearly detected the expression of the Piezo1-tdTomato proteins in the Piezo1-tdTomato-KI cardiomyocytes, but not in the control cells (Fig. 1b). We have previously found that Piezo1 localized in the plasma membrane interacts with the Sarco/Endoplasmic-Reticulum Calcium ATPase (SERCA)[35], which plays a key role in recycling cytosolic $Ca^{2+}$ back into the sarcoplasmic reticulum (SR) for maintaining SR $Ca^{2+}$ content of cardiomyocytes[36]. Indeed, co-immunostaining of

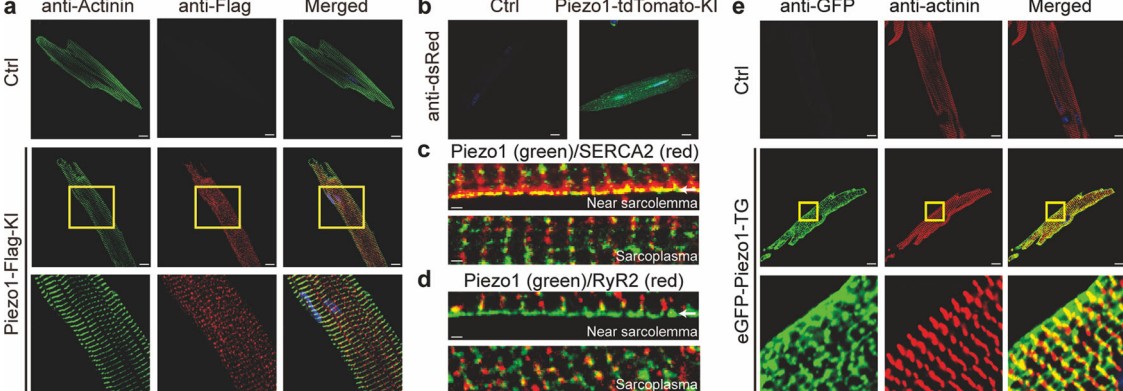

**Fig. 1 Expression and localization of Piezo1 in mouse cardiomyocytes. a** Immunofluorescent staining of adult cardiomyocytes isolated from either the control littermates (Ctrl) or Piezo1-Flag-KI mice. Green shows the anti-α-actinin signal for displaying the Z-lines of the striated cardiomyocytes, while red shows the anti-Flag signal for the endogenously expressed Flag-tagged Piezo1 protein. The bottom panel shows the enlarged section marked with the yellow box in the middle panel. Scale bar, 10 μm. **b** Immunofluorescent staining of adult cardiomyocytes isolated from either the control littermates (Ctrl) or the Piezo1-tdTomato-KI mice with the anti-dsRed antibody. Scale bar, 10 μm. **c, d** Co-immunofluorescent staining of the endogenously expressed Piezo1 with either SERCA2 (**c**) or RyR2 (**d**) in cardiomyocytes derived from the Piezo1-tdTomato-KI mice. The near plasma membrane and sarcoplasmic regions are respectively shown in the top and bottom panels. The Piezo1 signal is shown in green, while the signal of SERCA2 or RyR2 in red. The white arrow on the top panel indicates the sarcolemma region. **e** Immunofluorescent staining of adult cardiomyocytes isolated from either the control littermates (Ctrl) or the eGFP-Piezo1-TG mice. Green shows the anti-GFP signal for displaying the cardiac specific overexpression of the eGFP-Piezo1 fusion proteins, while red shows the anti-α-actinin signal. The bottom panel shows the enlarged section marked with the yellow box in the middle panel. Scale bar, 10 μm. Each experiment was repeated independently three times with similar results.

the endogenously expressed Piezo1-tdTomato and the type 2 SERCA (SERCA2) in cardiomyocytes revealed their colocalization near the sarcolemma but not inside the sarcoplasma (Fig. 1c). In contrast, sarcolemma-localized Piezo1 was not co-localized with the SR-localized type 2 ryanodine receptor (RyR2) (Fig. 1d), which is the intracellular $Ca^{2+}$ release channel responsible for the elementary $Ca^{2+}$ release, the so-called $Ca^{2+}$ spark, which underlies the cardiac excitation-contraction coupling[37,38]. Together, these data demonstrate that endogenous Piezo1 proteins are clearly expressed in the sarcolemma of adult cardiomyocytes.

We also generated cardiac-specific transgenic mice over-expressing the eGFP-Piezo1 fusion protein by crossing the MLC2v (myosin light chain 2 v)-Cre mice that exclusively express the Cre recombinase in ventricular cardiomyocytes[39] with the Piezo1-TG[fl-mCherry-stop-fl] mice, in which the upstream floxed mCherry coding sequence with a stop codon blocks the translation of the downstream eGFP-Piezo1 fusion protein[40] (Supplementary Fig. 2b). The resulting Piezo1-TG[fl-mCherry-stop-fl]/MLC2v-Cre (Piezo1-TG) mice had the floxed mCherry-stop codon sequence excised and accordingly expressed the eGFP-Piezo1 fusion proteins. Notably, the overexpressed eGFP-Piezo1 proteins showed similar puncate and striated localization patterns as that of the endogenously expressed Piezo1 (Fig. 1e).

**Piezo1 mediates Yoda1 induced $Ca^{2+}$ responses in cardiomyocytes.** To assay the functional role of the cardiac Piezo1, we generated cardiac-specific Piezo1-deficient mice (KO) by crossing the Piezo1[fl/fl] mice[24] with the MLC2v-Cre mice[39] (Supplementary Fig. 2a). RT-PCR and western blotting confirmed the successful deletion of Piezo1 in the KO heart (Fig. 2a, b). In contrast, the expression of Piezo1 in lung, blood vessel, and red blood cells was not affected (Fig. 2a and Supplementary Fig. 2c), suggesting specific deletion of Piezo1 in cardiomyocytes. Given that it is highly challenging to record stretch-activated currents from adult cardiomocytes[41], we have employed single-cell $Ca^{2+}$ imaging to functionally assay Piezo1 activity in cardiomyocytes in response to Yoda1, which is a chemical activator of Piezo1 with an apparent $EC_{50}$ (the concentration for causing half maximal activation) of ~17 μM and a maximal water solubility of ~30 μM[42]. We thus used 30 μM Yoda1 for maximal activation of endogenously expressed Piezo1 in cardiomyocytes. For comparison, we used 10 mM caffeine to activate RyR2 for measuring the total SR $Ca^{2+}$ content, which is a key determinant of cardiac $Ca^{2+}$ signaling and excitation-contraction coupling[11]. Yoda1 induced a sustained $Ca^{2+}$ response in cardiomyocytes derived from the littermate control mice in the presence of 1.8 mM extracellular $Ca^{2+}$ (Fig. 2c, d). Removing the extracellular $Ca^{2+}$ abolished the response, suggesting Yoda1-induced $Ca^{2+}$ influx (Fig. 2e). The amplitude of the $Ca^{2+}$ increase reached about 94% of the caffeine response (Fig. 2c, d, f, g), suggesting relatively robust Yoda1-induced $Ca^{2+}$ increase in cardiomyocytes. Remarkably, the Yoda1 response was nearly completely abolished in cardiomyocytes derived from the Piezo1-KO mice (Fig. 2c, d). On the other hand, cardiomyocytes derived from the Piezo1-TG mice had significantly larger Yoda1-induced $Ca^{2+}$ response than that detected in the littermate control cells (Fig. 2j, k). Unexpectedly, the caffeine-induced $Ca^{2+}$ release was significantly reduced in both Piezo1-KO (Fig. 2f, g) and Piezo1-TG (Fig. 2l) cardiomyocytes compared to their respective control cells, indicating decreased SR $Ca^{2+}$ content. To exclude the possibility that the diminished Yoda1-response in the KO cells might be due to their decreased SR $Ca^{2+}$ content, which might result in impaired $Ca^{2+}$-induced $Ca^{2+}$ release (CICR), we carried out total internal reflection fluorescent microscopy to image localized $Ca^{2+}$ influx near the plasma membrane. Remarkably, we detected rapid and robust

Yoda1-induced $Ca^{2+}$ events in control cardiomyocytes, which were nearly abolished in the KO cells (Fig. 2h, i). These data collectively demonstrate that Piezo1 mediates Yoda1-induced $Ca^{2+}$ responses in cardiomyocytes.

Notably, compared to caffeine-induced $Ca^{2+}$ response (Fig. 2f), the Yoda1-induced response in both control and Piezo1-TG cells appeared to be slower, but more sustained (Fig. 2c, j). These data suggest that the relatively slow global increase of cytosolic $Ca^{2+}$ evoked by Yoda1 was not caused by the expression level of Piezo1. Instead, we noticed that the onset of Yoda1-induced localized $Ca^{2+}$ influx (Fig. 2h) appeared to be much faster than Yoda1-induced global cytosolic $Ca^{2+}$ increase assayed by single-cell $Ca^{2+}$ imaging (Fig. 2c, j). Thus, we reasoned that the Yoda1-induced global cytosolic $Ca^{2+}$ increase might result from a summation of unsynchronized $Ca^{2+}$ influx mediated by Piezo1 distributed in the large sarcolemma area including T-tubules (Fig. 1), resulting in the relatively slow onset of the Yoda1 response observed in single-cell $Ca^{2+}$ imaging.

**Piezo1 mediates stretch-induced $Ca^{2+}$ sparks in cardiomyocytes.** Previous studies have shown that an 8% of diastolic stretch of ventricular myocytes can lead to a burst of $Ca^{2+}$ sparks[43,44]. To examine whether Piezo1 might mediate such stretch-induced $Ca^{2+}$ response, we have adopted the previously developed Myo-Tak system to precisely stretch a single cardiomyocyte and simultaneously measured the occurrence of $Ca^{2+}$ sparks using line-scan confocal microscopy[43]. Consistent with previous reports[43,44], an 8% stretch of the control cardiomyocytes evoked an apparent and rapid increase in $Ca^{2+}$ sparks, which was not observed in the Piezo1-KO cardiomyocytes (Fig. 3a, b). The fold change of $Ca^{2+}$ spark rate upon stretch in the control cardiomyocytes ($7.2 \pm 2.7$ folds) is significantly higher than that observed in the KO cardiomyocytes ($1.4 \pm 0.2$ folds) (Fig. 3c). The stretch-induced $Ca^{2+}$ sparks were largely recovered upon relaxation from the stretch (Fig. 3c). To further demonstrate stretch dependence of $Ca^{2+}$ sparks, we applied a 15% of stretch that is within the range of a diastolic stretch normally occurred during a healthy heart beat. Compared to the 8% stretch-induced response, 15% stretch induced a relatively more robust increase in $Ca^{2+}$ sparks, which could still largely recover upon relaxation (Fig. 3d, e), indicating no major damage to the cells. Importantly, such responses were not observed in the Piezo1-KO cardiomyocytes (Fig. 3d, e). To exclude the possibility that the lack of stretch-induced $Ca^{2+}$ spark in the KO cells might be due to their decreased SR $Ca^{2+}$ content, we employed 1 mM caffeine to trigger $Ca^{2+}$ sparks (Fig. 3f, g). Despite relatively lower $Ca^{2+}$ spark events in the KO cardiomyocytes due to their reduced SR $Ca^{2+}$ content, the caffeine-induced fold change of $Ca^{2+}$ sparks were similar between the control and KO cells (Fig. 3g). These data suggest that the Piezo1-KO cardiomyocytes specifically lost stretch-induced but retained caffeine-induced $Ca^{2+}$ sparks. Taken together, these data demonstrate that Piezo1 mediates stretch-induced $Ca^{2+}$ sparks in cardiomyocytes.

**Piezo1 affects $Ca^{2+}$ homeostasis in cardiomyocytes.** The observation that Piezo1-KO cardiomyocytes had both reduced SR $Ca^{2+}$ content and less spontaneous $Ca^{2+}$ sparks prior to stretch stimulation has prompted us to further examine the role of Piezo1 in controlling cardiac $Ca^{2+}$ homeostasis. Indeed, a direct comparison of the spontaneous $Ca^{2+}$ sparks between the control and KO cardiomyocytes revealed significantly less frequent $Ca^{2+}$ spark events in the KO cells than in the control cells ($0.5 \pm 0.6$ vs $2.0 \pm 0.2$ events per 10 μm per second, respectively) (Fig. 4a, c). On the basis of histogram analysis, the peak amplitude of the $Ca^{2+}$ spark from the KO cells was slightly lower than that from

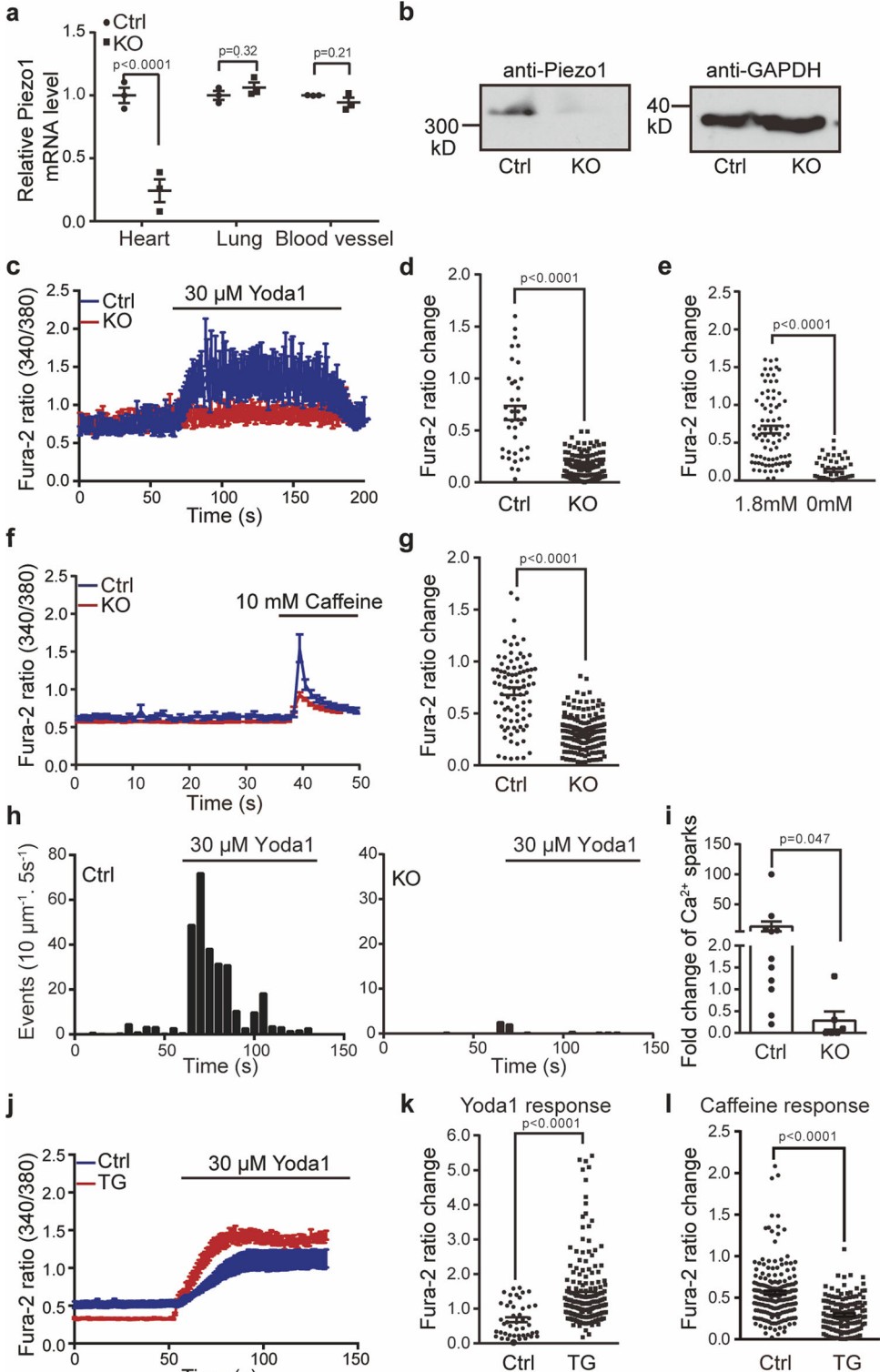

the control cells ($1.65 \pm 0.03$ vs $1.82 \pm 0.03$) (Fig. 4b). The decrease in SR Ca$^{2+}$ content and spontaneous Ca$^{2+}$ spark in the KO cells indicates that Piezo1-mediated Ca$^{2+}$ influx might help to maintain the SR Ca$^{2+}$ store.

On the other hand, despite decreased SR Ca$^{2+}$ content, the Piezo1-TG cardiomyocytes showed significantly enhanced Ca$^{2+}$ spark events compared to the control cells ($3.6 \pm 0.7$ vs $1.2 \pm 0.2$ events per 10 μm per second, respectively) (Fig. 4d, f).

Furthermore, the peak amplitude of the Ca$^{2+}$ spark from the TG cells is drastically lower than that of the control cells ($1.03 \pm 0.03$ vs $1.74 \pm 0.02$, respectively) (Fig. 4e). In addition to the frequently occurred local Ca$^{2+}$ sparks, about 80% of the TG cells showed spontaneous Ca$^{2+}$ waves (Fig. 4g, h), which represent transient but global increase in the intracellular Ca$^{2+}$ that propagate as waves through the cell and are highly arrhythmogenic[45]. In comparison, less than 30% of the control cells showed such Ca$^{2+}$

**Fig. 2 Piezo1-dependent Ca$^{2+}$ response in mouse cardiomyocytes. a** Scatter plot of relative *Piezo1* mRNA level from the littermate control (Ctrl) ($n=3$) and cardiac-specific Piezo1-KO tissues ($n=3$). Unpaired student's t-test, two-sided. Values are mean ± SEM. **b** Representative western blotting result of Piezo1 proteins immunoprecipitated from Ctrl and KO heart homogenates using the anti-Piezo1 antibody. The GAPDH level was used for loading control. Experiment was repeated independently three times with similar results. **c** Representative average traces of single-cell Fura-2 Ca$^{2+}$ imaging of Ctrl (7 cells) or KO (8 cells) cardiomyocytes in response to the Piezo1 chemical activator Yoda1. **d** Scatter plot of Yoda1-induced Fura-2 amplitude change of Ctrl (39 cells) and KO (127 cells) cardiomyocytes from control ($n=3$) mice and their littermate KO ($n=4$) mice. Unpaired student's t-test, two-sided. **e** Scatter plot of Yoda1-induced Fura-2 amplitude of cardiomyocytes with (84 cells) or without (49 cells) 1.8 mM extracellular Ca$^{2+}$ from 3 mice. Unpaired student's t-test, two-sided. **f** Representative average traces of single-cell Fura-2 Ca$^{2+}$ imaging of Ctrl (13 cells) and KO (20 cells) cardiomyocytes in response to 10 mM caffeine. **g** Scatter plot of caffeine-induced Fura-2 ratio changes from the indicated cardiomyocytes, reflecting the SR Ca$^{2+}$ store level. Ctrl (89 cells) and KO (151 cells) from KO ($n=3$) mice and their littermate control ($n=3$) mice. Unpaired student's t-test, two-sided. **h** Ca$^{2+}$ spark histogram before and after application of Yoda1 of Ctrl (11 cells) and KO (6 cells) cardiomyocytes by TIRF. Each column represents the average of Ca$^{2+}$ sparks per cell in every 5 s. **i** Scatter plot of fold-change of Ca$^{2+}$ spark rate in Ctrl (11 cells) or KO (6 cells) cardiomyocytes from KO ($n=3$) mice and their littermate control ($n=3$) mice during Yoda1-treatment. **j** Representative average traces of single-cell Fura-2 Ca$^{2+}$ imaging of Ctrl (11 cells) or TG (20 cells) cardiomyocytes in response to Yoda1. **k** Scatter plot of Yoda1-induced Fura-2 amplitude change of Ctrl (45 cells) and TG (191 cells) cardiomyocytes from TG ($n=4$) mice and their littermate control ($n=3$) mice. Unpaired student's t-test, two-sided. **l** Scatter plot of caffeine-induced Fura-2 ratio changes from the indicated cardiomyocytes, reflecting the SR Ca$^{2+}$ store level. Ctrl (201 cells) and TG (137 cells) from TG ($n=3$) mice and their littermate control ($n=3$) mice. Unpaired student's t-test, two-sided. Values are mean ± SEM in **c–g**, **i**, **k–l**.

waves (Fig. 4h). The occurring frequency of the Ca$^{2+}$ waves in the TG cells is also much higher than that in the control cells (4.6 ± 1.6 vs 0.7 ± 0.2 per 100 μm per second, respectively) (Fig. 4i). Given the markedly increased resting Ca$^{2+}$ sparks and waves in the TG cells, their decreased SR Ca$^{2+}$ content is highly likely due to profound Ca$^{2+}$ leakage from the SR Ca$^{2+}$ store. Collectively, these data demonstrate that Piezo1 has a prominent role in determining Ca$^{2+}$ homeostasis in cardiomyocytes.

**Piezo1 mediates stretch-induced and homeostatic ROS signaling.** We next set up to understand how Piezo1 might affect the stretch-induced and homeostatic Ca$^{2+}$ signaling in cardiomyocytes. On the one hand, Piezo1-mediated Ca$^{2+}$ influx might directly contribute to cardiac Ca$^{2+}$ homeostasis via the CICR mechanism. On the other hand, previous studies have suggested that stretch-induced Ca$^{2+}$ sparks are mediated by a mechano-chemo signaling pathway termed X-ROS signaling, which involves stretch-activation of nicotinamide adenine dinucleotide phosphate (NADPH) oxidase 2 (NOX2) via Rac1-dependent activation of microtubules, leading to the production of reactive oxygen species (ROS) that in turn modulates the activity of RyR2[44]. However, whether mechanosensitive ion channels are involved in the X-ROS signaling has remained unclear. Using GsMTx4 as a blocker for mechanosensitive ion channels, one study has shown that GsMTx4 had no effect on the stretch-induced acute increase in Ca$^{2+}$ spark rate[43], while another study has observed blocking effect[46].

We, therefore, asked whether Piezo1 might be involved in this X-ROS signaling pathway by using the Piezo1-KO mouse model. To test this hypothesis, we measured ROS production in cardiomyocytes using the fluorescent ROS sensor 2′,7′-dichloro-fluorescein diacetate (DCF). Interestingly, Yoda1 induced a significant increase in DCF fluorescence in control cardiomyocytes in the presence of extracellular Ca$^{2+}$ (Fig. 5a, b). In contrast, in the absence of extracellular Ca$^{2+}$, the Yoda1-induced increase in DCF fluorescence was totally abolished (Fig. 5c, d). Remarkably, Piezo1-KO cardiomyocytes also totally lost the Yoda1-induced DCF response (Fig. 5e, f). Given that Yoda1 can induce Piezo1-dependent Ca$^{2+}$ influx into cardiomyocytes (Fig. 2c–e), these data suggest that Yoda1-induced increase in ROS production is mediated by Piezo1-dependent Ca$^{2+}$ influx.

We next directly tested the requirement of Piezo1 for stretch-induced ROS production. In line with previous reports[44], we observed that an 8% of stretch induced an apparent increase in DCF intensity in the control cardiomyocytes (Fig. 5g, h). Remarkably, such response was nearly completely abolished in the Piezo1-KO cardiomyocytes (Fig. 5g, h). These data provide compelling genetic evidence that Piezo1 mediates both Yoda1- and stretch-induced production of ROS in cardiomyocytes. Despite that GsMTx4 is able to block heterologously expressed Piezo1, its effect on endogenously expressed Piezo1 in cardio-myocytes might be variable, which might account for the inconsistent results of the previous studies[43,46].

Taking advantage of Yoda1-induced ROS production, we next went on to ask whether Piezo1 might actually serve as the direct upstream mechanotransducer of the stretch-induced X-ROS signaling pathway by providing Ca$^{2+}$ influx (Fig. 5i). If this is the case, we reasoned that Yoda1-induced ROS production would be prevented by blocking the downstream Rac1 and NOX2 (Fig. 5i). Indeed, blocking the Ca$^{2+}$-dependent small GTPase Rac1 with an inhibitor not only prevented Yoda1-induced ROS production as indicated by the lack of DCF fluorescence increase, but also significantly suppressed the basal level of ROS (Fig. 5j), suggesting a constitutive Rac1-dependent ROS production. Furthermore, pre-incubation of the membrane-permeable NOX2 inhibiting peptide gp91ds-tat blocked Yoda1-induced production of ROS (Fig. 5k). Together, these data demonstrate that Piezo1-serves as the upstream mechanotransduction channel to initiate the Ca$^{2+}$ influx-Rac1-NOX2-ROS-Ca$^{2+}$ spark signaling pathway (Fig. 5i).

Interestingly, in the absence of mechanical or Yoda1 stimulation, compared to the control cardiomyocytes, the Piezo1-KO cells had significantly reduced DCF fluorescence (Fig. 5l), while the Piezo1-TG cells showed significantly enhanced DCF fluorescence (Fig. 5m), demonstrating that Piezo1 controls the ROS home-ostasis in a dose-dependent manner.

**Cardiac-specific deletion of Piezo1 impairs heart function.** Ca$^{2+}$ is essential for the cardiac excitation-contraction coupling[11]. Abnormal Ca$^{2+}$ handling leads to heart dysfunction[45]. Given the critical role of Piezo1 in mediating cardiac Ca$^{2+}$ signaling, we next examined the impact of cardiac-specific deletion of Piezo1 on heart function. When examined at 8-week old, the cardiac-specific Piezo1-KO mice did not display obvious defects in heart and body weights, heart morphology, and pump function (Supplementary Fig. 3a–e and Supplementary Table 1a). Furthermore, echo-cardiographic analysis revealed no apparent structural defects in the left ventricles of the KO mice (Supplementary Movies 1–4). Thus, the Piezo1-KO mice at 8-week old showed overall normal heart structure and function, suggesting that the MLC2v-Cre dependent deletion of Piezo1 might not cause developmental defects of the heart during the embryonic developmental stage.

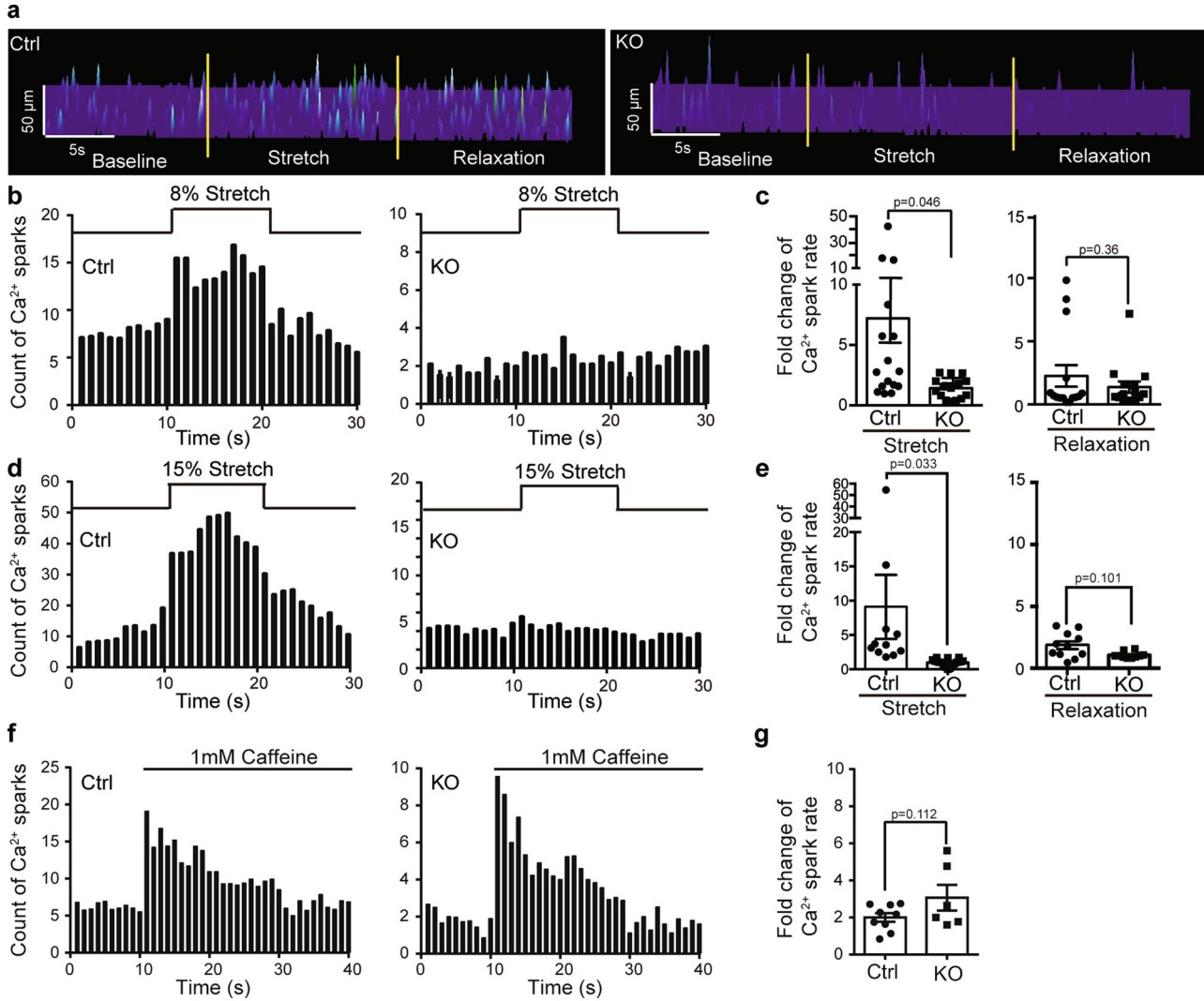

**Fig. 3 Piezo1 mediates stretch-induced Ca²⁺ signaling.** **a** Representative fluorescence surface plot of stretch-activated $Ca^{2+}$ sparks in Ctrl or KO cardiomyocytes in response to an 8% axial stretch. $Ca^{2+}$ sparks were detected using line-scan mode of confocal microscopy of the $Ca^{2+}$ indicator Fluo-4 AM. **b** $Ca^{2+}$ spark histogram before, during, and after 8% axial stretch of Ctrl (16 cells) and KO (17 cells) cardiomyocytes from KO ($n = 3$) mice and their littermate control ($n = 3$) mice. **c** Scatter plot of fold-change of $Ca^{2+}$ spark rate in Ctrl (16 cells) or KO (17 cells) cardiomyocytes from KO ($n = 3$) mice and their littermate control ($n = 3$) mice during stretch (left panel) and upon relaxation (right panel). Unpaired student's t-test, two-sided. Values are mean ± SEM. **d** $Ca^{2+}$ spark histogram before, during, and after 15% axial stretch of Ctrl (11 cells) and KO (20 cells) cardiomyocytes from KO ($n = 3$) mice and their littermate control ($n = 3$) mice. **e** Scatter plot of fold-change of $Ca^{2+}$ spark rate in Ctrl (11 cells) or KO (20 cells) cardiomyocytes from KO ($n = 3$) mice and their littermate control ($n = 3$) mice during stretch (left panel) and upon relaxation (right panel). Unpaired student's t-test, two-sided. Values are mean ± SEM. **f** $Ca^{2+}$ spark histogram before and during the application of 1 mM caffeine of Ctrl (9 cells) and KO (6 cells) cardiomyocytes from KO ($n = 3$) mice and their littermate control ($n = 3$) mice. **g** Scatter plot of fold-change of $Ca^{2+}$ spark rate in Ctrl (9 cells) or KO (6 cells) cardiomyocytes from KO ($n = 3$) mice and their littermate control ($n = 3$) mice during application of caffeine. Unpaired student's t-test, two-sided. Values are mean ± SEM.

At 18-week old, the KO mice unexpectedly had an increased body weight, but with their heart weight and the ratio of heart weight to body weight comparable to the littermate control mice (Fig. 6a–c). Histological analysis revealed that the KO hearts appeared moderately larger with dilated left ventricles (Fig. 6d). Masson's trichrome staining of the left ventricle sections revealed significantly enhanced intermuscular fibrosis in the KO heart sections, indicating the development of cardiomyopathy (Fig. 6e, f). Indeed, RT-PCR of the KO heart tissues verified significantly increased mRNA levels of cardiomyopathic marker genes including *β-myosin heavy chain* (*MHC*) and *atrial natriuretic peptide* (*ANP*) (Fig. 6g). ECG examination did not show obvious cardiac arrhythmias in the control and KO mice (Fig. 6h). However, echocardiography showed that the KO heart had a significantly

increased end-diastolic and end-systolic internal diameter and volume of left ventricle (Fig. 6i, j and Supplementary Table 1b). Both the diastolic and systolic volume of the left ventricle of the KO hearts was significantly enlarged, resulting in unchanged stroke volume (Fig. 6k and Supplementary Table 1b). The observation that a stretched diastolic left ventricle of the KO hearts (81.8 ± 8.7 μl) relative to that of the control hearts (61.2 ± 3.4 μl) led to a comparable stroke volume suggests a compromised stoke function and Frank–Starling response in the KO heart. The KO hearts had a concomitant decreased ejection fraction and fractional shortening (Fig. 6l, m and Supplementary Table 1b), demonstrating impaired heart pump function. Thus, in line with its prominent role in regulating cardiac $Ca^{2+}$ handling and ROS signaling, Piezo1 plays a critical role in maintaining normal heart functions.

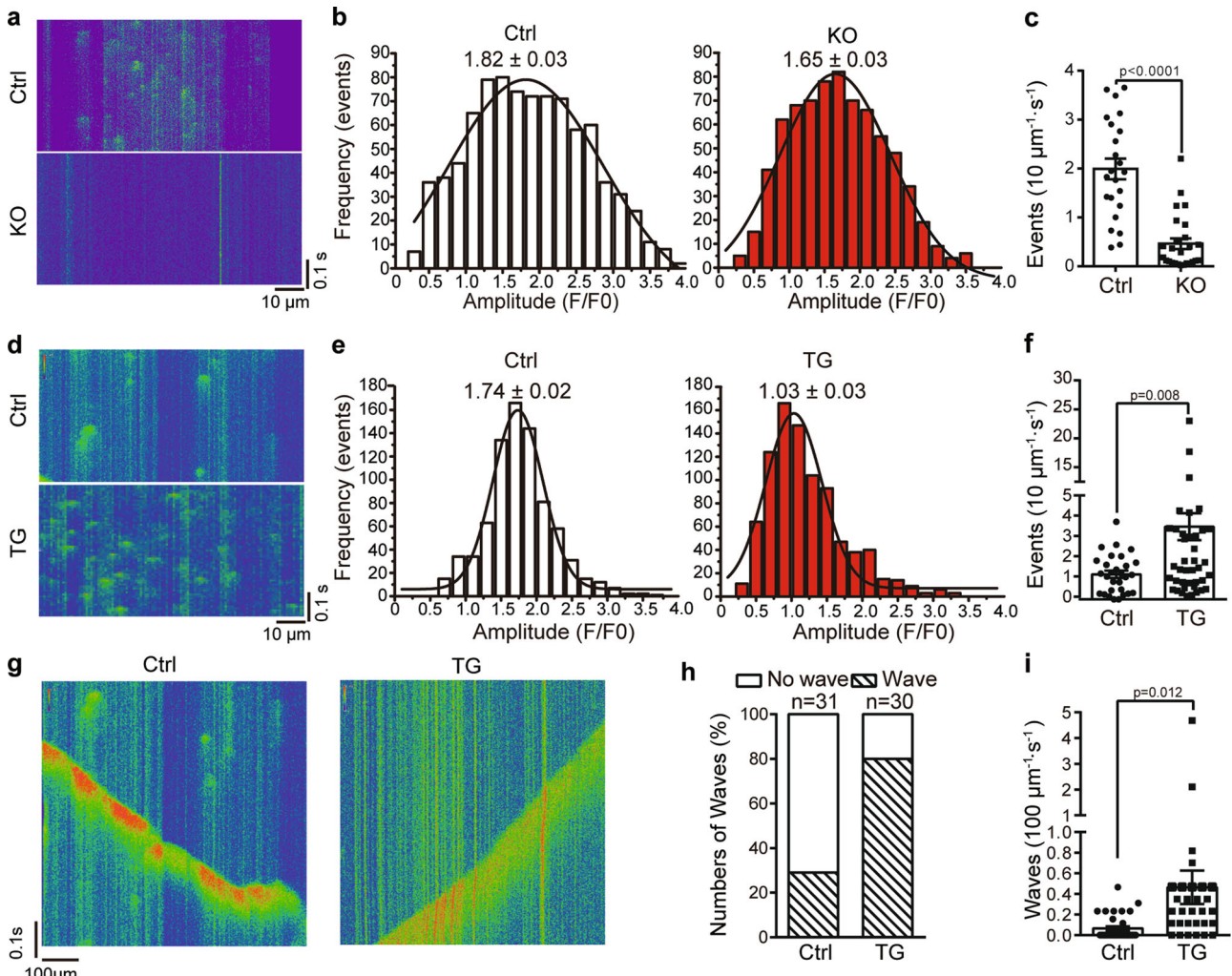

**Fig. 4 Piezo1 mediates homeostatic Ca²⁺ signaling. a**, **d** Representative fluorescence surface plot of spontaneous Ca²⁺ sparks of the indicated cardiomyocytes. **b**, **e** Histogram analysis of the Ca²⁺ spark frequency of the indicated cardiomyocytes. The peak amplitudes are labeled above the fit. **c**, **f** Scatter plot of Ca²⁺ spark of cardiomyocytes with the indicated genotypes [23 Ctrl cells and 26 KO cells from KO ($n = 5$) mice and their littermate control ($n = 5$) mice in **b**, **c** 29 Ctrl cells and 47 TG cells from TG ($n = 3$) mice and their littermate control ($n = 3$) mice in **e**, **f**]. Unpaired student's t-test, two-sided. Values are mean ± SEM. **g** Representative fluorescence surface plot of spontaneous Ca²⁺ waves of the indicated cardiomyocytes. **h** Proportion of the indicated cardiomyocytes with or without Ca²⁺ waves [31 Ctrl cells and 30 TG cells from TG ($n = 3$) mice and their littermate control ($n = 3$) mice]. **i** Scatter plot of frequency of Ca²⁺ waves of the indicated cardiomyocytes [31 Ctrl cells and 30 TG cells from TG ($n = 3$) mice and their littermate control ($n = 3$) mice]. Unpaired student's t-test, two-sided. Values are mean ± SEM.

**Cardiac-specific overexpression of Piezo1 induces heart failure and arrhythmias**. Given the markedly increased occurrence of Ca²⁺ sparks and waves and decreased SR Ca²⁺ store upon overexpression of Piezo1 in cardiomyocytes, we expected abnormal heart function in the cardiac-specific Piezo1-TG mice. Remarkably, compared to littermate control mice, the TG mice at 8-week old showed significantly enlarged hearts with apparently dilated chambers without thickening of the ventricle wall (Fig. 7a–d), suggesting dilated cardiomyopathy. The morphological change of the TG heart progressed more severely with increased age. At 18-week old, the heart weight of the TG mice was nearly twice of the control heart weight (457 ± 46 mg vs 223 ± 7 mg, respectively) (Fig. 7b). The TG mice had significantly lower body weight than the control mice at both 8-week and 18-week old (Fig. 7c). The heart weight to body weight ratio of the TG mice at 18-week old reached about 3 folds of that of the control mice (17.1 ± 2.3 vs 5.7 ± 0.2) (Fig. 7d). In contrast. there was no significant difference in tibia length between the TG and control mice (Fig. 7e), indicating that the growth and size of the

TG mice was normal. The TG heart tissues also showed significantly enhanced intermuscular fibrosis (Fig. 7f, g) and increased mRNA levels of $\beta$-MHC and ANP (Fig. 7j). Remarkably, while none of the 12 littermate control mice showed arrhythmia (Fig. 7h), all the 12 TG mice examined at 4-week, 8-week or 18-week old showed ventricular tachycardia (Fig. 7h), in line with the frequently observed arrhythmogenic Ca²⁺ waves in the TG cardiomyocytes (Fig. 4l–n). Echocardiographic analysis of 4-week, 8-week and 18-week old mice (Supplementary Table 1c–e) showed that the TG mice had severely impaired heart pump function as reflected by their significantly increased internal diameter and volume of left ventricle (Fig. 7k–n) and a markedly decreased ejection fraction and fractional shortening (Fig. 7p, q). However, echocardiographic analysis revealed no structural defects in the left ventricle of TG mice (Supplementary Movies 5–8), indicating that the defective pump function of the TG mice might not be due to developmental defects. Together, these characterizations demonstrate that cardiac-specific overexpression of Piezo1 leads to severe heart failure and arrhythmias,

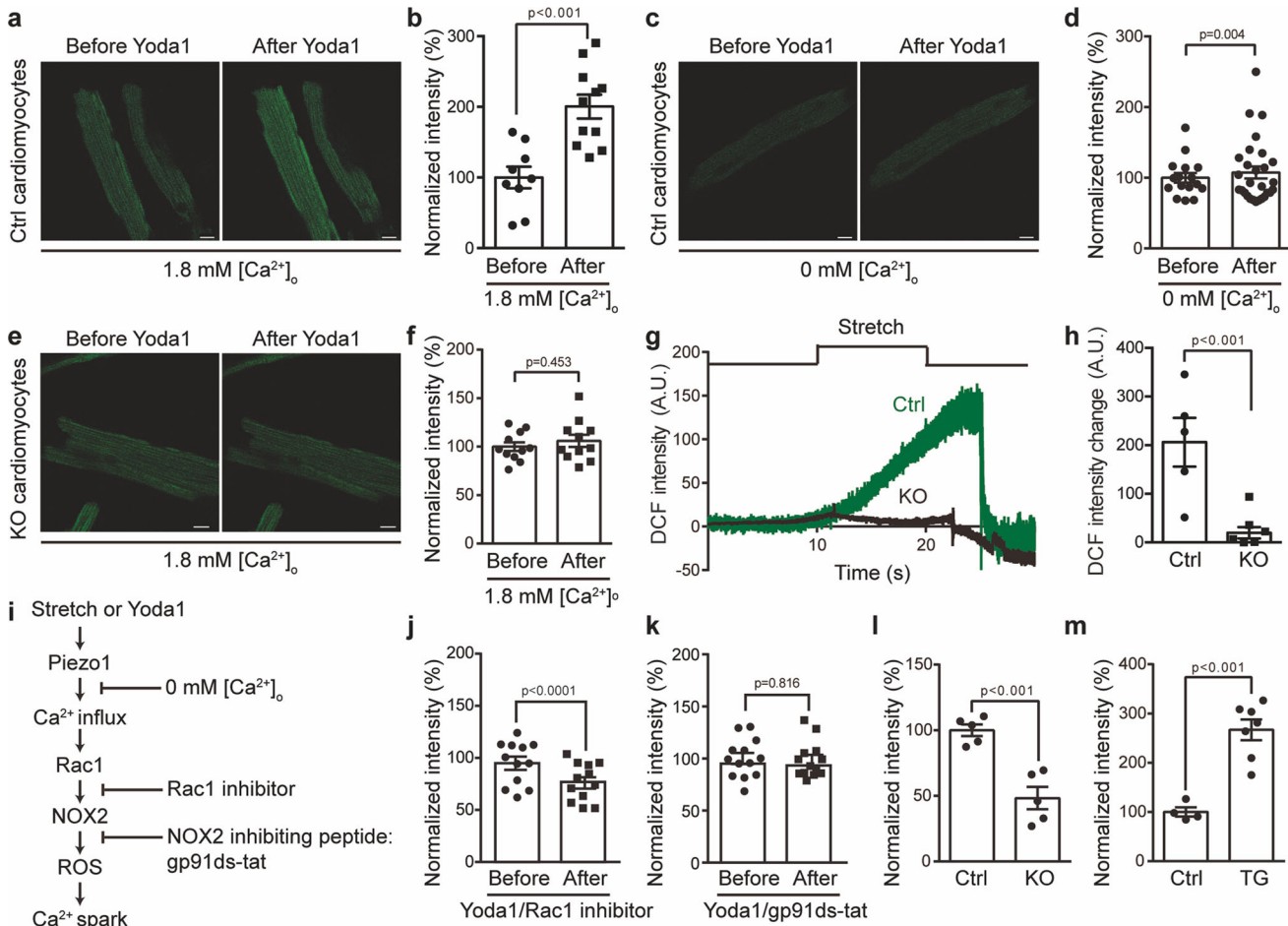

**Fig. 5 Piezo1 mediates stretch-induced and homeostatic ROS signaling. a, c** Representative DCF fluorescent images of Ctrl cardiomyocytes before and after application of Yoda1 in the presence (**a**) or absence (**c**) of 1.8 mM $[Ca^{2+}]_o$. Scale bar, 10 μm. **b, d** Scatter plot of normalized DCF intensity of Ctrl cardiomyocytes before and after application of Yoda1 in the presence (11 cells) or absence (15 cells) of 1.8 mM $[Ca^{2+}]_o$. 3 mice for each group. Paired student's t-test, two-sided. **e** Representative DCF fluorescent images of KO cardiomyocytes before and after application of Yoda1 in the presence of 1.8 mM $[Ca^{2+}]_o$. Scale bar, 10 μm. **f** Scatter plot of normalized DCF intensity of KO cardiomyocytes (11 cells from 3 KO mice) before and after application of Yoda1 in the presence of 1.8 mM $[Ca^{2+}]_o$. Paired student's t-test, two-sided. **g** Representative DCF fluorescent imaging trace of Ctrl or KO cardiomyocytes in response to 8% axial stretch. **h** Scatter plot of stretch-induced DCF fluorescence intensity change of Ctrl (5 cells) or KO cardiomyocytes (6 cells). 3 mice for each group. Unpaired student's t-test, two-sided. **i** The proposed scheme of Piezo1-mediated ROS-generating pathway. **j** Scatter plot of normalized DCF intensity of wild-type cardiomyocytes (12 cells from 3 mice of three independent experiments) in response to Yoda1 together with the Rac1 inhibitor. Paired student's t-test, two-sided. **k** Scatter plot of normalized DCF intensity of wild-type cardiomyocytes (14 cells from 3 mice of three independent experiment) in response to Yoda1 after pre-incubation with the membrane-permeable NOX2 inhibiting peptide gp91ds-tat. Paired student's t-test, two-sided. **l** Scatter plot of normalized DCF intensity of Ctrl (5 cells) and KO (5 cells) cardiomyocytes for three independent experiments. Unpaired student's t-test, two-sided. **m** Scatter plot of normalized DCF intensity of Ctrl (4 cells) and TG (7 cells) cardiomyocytes from three independent experiments. Unpaired student's t-test, two-sided. Values are mean ± SEM in **b, d, f, h, j–m**.

consistent with abnormal $Ca^{2+}$ handling and ROS signaling in the Piezo1-TG cardiomyocytes.

**Autonomic upregulation of Piezo1 contributes to the development of cardiomyopathy.** Given that both deletion and overexpression of Piezo1 can lead to heart dysfunction in mice (Figs. 6, 7), we wondered the pathological contribution of Piezo1 to cardiomyopathy. A previous study has suggested upregulation of Piezo1 in myocardial infarction-induced failing rat hearts[34]. Our studies of the mechanotransduction of osteoblasts have revealed a positive feedback loop between mechanical force and the Piezo1 mechanosensor itself, in which a mechanical load leads to increased expression of Piezo1[47]. To examine whether heart might possess a similar positive feedback mechanism for regulating Piezo1 expression in response to mechanical stress such as during the development of cardiomyopathy, we subjected the Piezo1-Flag-KI mice for

doxorubicin-induced dilated cardiomyopathy[48]. Western blotting of the anti-Flag-immunoprecipitated samples from cardiomyocytes derived from doxorubicin-treated Piezo1-Flag-KI mice revealed more abundant Piezo1 proteins than that from the saline-treated Piezo1-Flag-KI mice (Fig. 8a). The sample from saline-treated wild-type control mice was used as a negative control for demonstrating the Piezo1-specific western blotting signal (Fig. 8a). Consistent with upregulation of Piezo1 proteins in diseased mouse and rat hearts (Fig. 8a)[34], RT-PCR revealed a 5-fold increase in the mRNA expression of *Piezo1* in human heart samples with hypertrophic cardiomyopathy compared to normal heart samples (Fig. 8b and Supplementary Excel Sheet). Thus, Piezo1 is upregulated under pathological conditions.

To test whether the upregulation of Piezo1 expression in diseased hearts might be due to an autonomic response of cardiomyocytes, we subjected cardiomyocytes derived from the

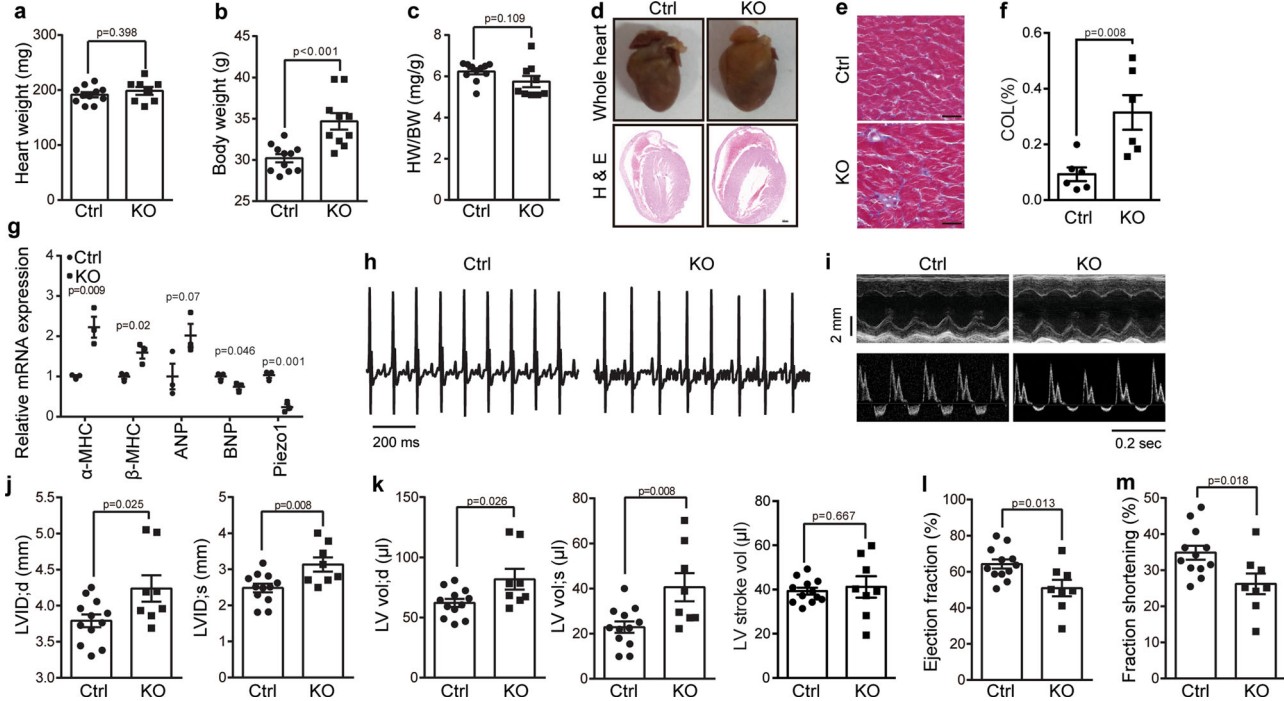

**Fig. 6 Cardiac specific knockout of Piezo1 impairs heart function. a–c** Scatter plot of heart weight (**a**), body weight (**b**), and HW (heart weight)/BW (body weight) ratio (**c**) of 18-week old littermate Ctrl ($n = 11$) and Piezo1-KO ($n = 10$) male mice. Unpaired student's t-test, two-sided. Values are mean ± SEM. **d** Histologic analysis of whole hearts or H & E stained longitudinal heart sections derived from 18-week old Ctrl and KO male mice. **e** Histologic analysis of the left ventricles of the Ctrl and KO hearts sectioned longitudinally and subjected to Masson's trichrome staining. Scale bar, 20 μm. **f** Scatter plot of collagen content in the ventricular area of the Ctrl and KO hearts derived from 18-week old mice ($n = 6$). Unpaired student's t-test, two-sided. Values are mean ± SEM. **g** RT-qPCR analysis of the normalized mRNA level of the indicated cardiac genes derived from18-week old Ctrl and KO mice ($n = 3$). Unpaired student's t-test, two-sided. Values are mean ± SEM. **h** Representative ECG recordings from 18-week old Ctrl and KO male mice. **i** Top panel: M-mode parasternal short-axis at mid-ventricular level of end-diastolic and end-systolic dimension of left ventricle; Lower panel: Pulsed Doppler at mitral valve inflow showing mitral blood flow velocity of early-diastolic and end-diastolic. **j–m** Scatter plots of echocardiographic analysis of left ventricular end-diastolic internal diameter (LVID;d) (left panel) and end-systolic internal diameter (LVID;s) (right panel) (**j**); diastolic (left panel) systolic (middle panel), and stroke volume (right panel) of left ventricle (**k**); percentage of ejection fraction (**l**); percentage of fractional shortening (**m**) determined from transthoracic M-mode tracings from 18-week old Ctrl ($n = 12$) and KO ($n = 8$) mice. Unpaired student's t-test, two-sided. Values are mean ± SEM. All experiments were conducted with male mice.

littermate control and the Piezo1-KO neonatal hearts to the hypertrophic agonist phenylephrine (PE) for modeling cellular hypertrophy[49]. PE-treated control myocytes showed increased expression of ANP (Fig. 8c), verifying the hypertrophic effect of PE. Consistent with the observed upregulation of Piezo1 in diseased heart tissues (Fig. 8a, b), the mRNA expression of Piezo1 was also significantly increased in PE-treated control myocytes compared to non-treated control cells (Fig. 8c), suggesting an autonomic upregulation of Piezo1 in cardiomyocytes. Intriguingly, while the KO myocytes without PE treatment showed higher expression of ANP and β-MHC than control myocytes without PE treatment, PE-treated KO cells unexpectedly had reduced expression of ANP and β-MHC (Fig. 8c). These data indicate that deletion of Piezo1 appears to reverse PE-induced hypertrophy of cardiomyocytes. Together with the observation that cardiac-specific Piezo1-TG mice displayed severe heart failure (Fig. 7), these data collectively demonstrate that maladaptive upregulation of Piezo1 in response to cardiac perturbation might contribute to the pathogenesis of cardiomyopathy in both mouse and human hearts.

## Discussion
Piezo1 is a bona fide mechanosensitive cation channel that utilizes its unique three-bladed, propeller-like architecture to effectively convert distinct forms of mechanical stimuli into $Ca^{2+}$ signaling in various cell types[14,50], and consequently plays critical

roles in various aspects of vascular physiology, including blood and lymphatic vessel development, vascular tone, arterial remodeling, and baroreflex control of blood pressure and heart rate[16,18,27]. Cardiomyocytes experience drastic mechanical changes on a beat-to-beat basis[1], and their mechanical responses are believed to underlie the well-known Frank–Starling law and Anrep effect that constitute powerful mechanisms to allow the heart to adapt to an abrupt rise in either preload or afterload[4]. Stretch-induced increase in intracellular $Ca^{2+}$ underlies a fundamental mechanism for stretch-dependent change in the development of mechanical force of cardiomyocytes. Here we have established that the mechanosensitive Piezo1 channel serves as a key cardiac mechanotransducer that directly converts mechanical stretch of cardiomyocytes into $Ca^{2+}$ and ROS signaling (Figs. 1–5), providing a crucial molecular basis underlying the positive relationship between mechanical stress and cardiac force production.

The observation that Piezo1 controls both stretch-induced and homeostatic $Ca^{2+}$ and ROS signaling might suggest a Piezo1-mediated positive feedback loop between these two signaling molecules critical for cardiac function. Piezo1-dependent $Ca^{2+}$ influx regulates the production and homeostasis of ROS via the Rac1-NOX2 signaling pathway, which might, in turn, sensitize $Ca^{2+}$ release via acting on RyR2, whose sensitivity can be modulated through ROS-mediated posttranslational modifications[51,52]. Such a positive feedback signaling transduction mechanism might

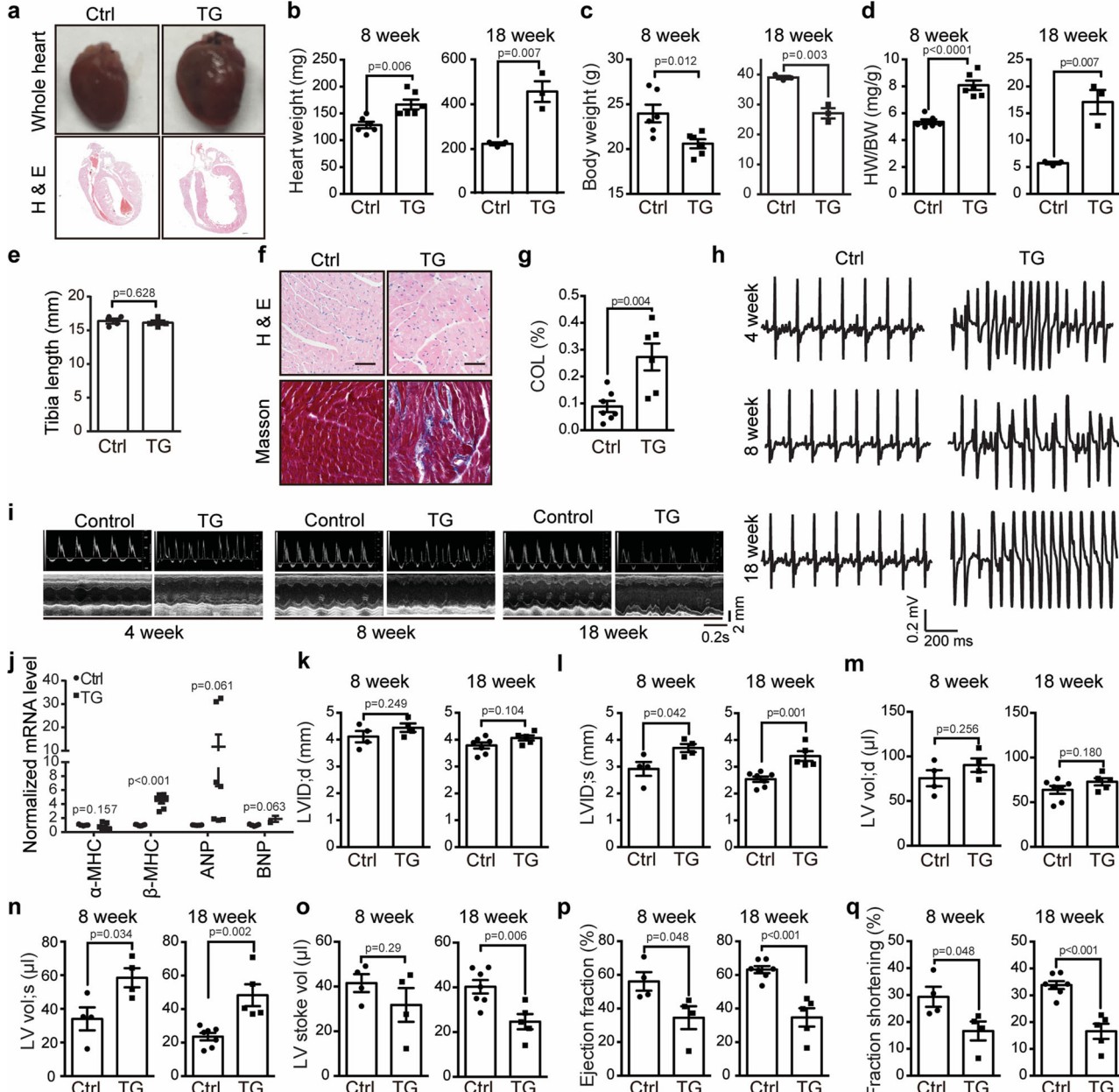

**Fig. 7 Cardiac-specific overexpression of Piezo1 induces severe heart failure and arrhythmias. a** Histologic analysis of whole hearts or H & E stained longitudinal heart sections derived from 8-week old littermate control (Ctrl) and Piezo1-TG male mice (TG). **b–d** Scatter plots of heart weight (**b**), body weight (**c**), and HW/BW ratio (**d**) of 8-week-old ($n = 6$ mice for each group) or 18-week old ($n = 3$ mice for each group) Ctrl and TG male mice. Unpaired student's t-test, two-sided. Values are mean ± SEM. **e** Scatter plots of tibia length of 8-week old ($n = 4$ mice for each group) Ctrl littermates and TG male mice. Unpaired student's t-test, two-sided. Values are mean ± SEM. **f** Histologic analysis of the left ventricles of 8-week old Ctrl and TG hearts sectioned longitudinally and subjected to either H & E (top panel) or Masson's trichrome staining (lower panel). Scale bar, 20 μm. **g** Scatter plot of collagen content in the ventricular area of 8-week old Ctrl and TG hearts ($n = 6$ for each group). Unpaired student's t-test, two-sided. Values are mean ± SEM. **h** Representative ECG recordings from 4-week, 8-week, or 18-week old Ctrl and TG male mice. **i** Representative echocardiographs of 4-week, 8-week, and 18-week old Ctrl and TG littermates. **j** RT-PCR analysis of the normalized mRNA level of the indicated cardiac genes derived from the Ctrl and TG mice ($n = 7$). Unpaired student's t-test, two-sided. Values are mean ± SEM. **k–q** Scatter plots of echocardiographic analysis of the indicated parameters from 8-week ($n = 4$ mice for each group) or 18-week ($n = 4$ mice for each group) Ctrl and TG mice. Unpaired student's t-test, two-sided. Values are mean ± SEM. All experiments were conducted with male mice.

allow a relatively low abundant but well sarcolemma-distributed Piezo1 to optimally respond to the drastic and repeated mechanical stress generated by the beating heart. Indeed, either deletion or overexpression of Piezo1 resulted in dysregulated $Ca^{2+}$ and ROS signaling and heart dysfunction (Figs. 6, 7). Piezo1-mediated $Ca^{2+}$ influx might also contribute to maintain the SR $Ca^{2+}$ store. Thus, we propose that the Piezo1-mediated mechano-chemo transduction

process might safeguard a homeostatic functional state of the heart. Disrupting such a homeostatic role might cause abnormal contractile function and arrhythmias. For instance, breaking the positive feedback loop upon Piezo1 deletion leads to a decreased $Ca^{2+}$ influx and ROS production, resulting in a combination of reduced SR $Ca^{2+}$ content (Fig. 2f, g) and unsensitized SR $Ca^{2+}$ release (Fig. 4a–c), which is expected to cause comprised heart pump

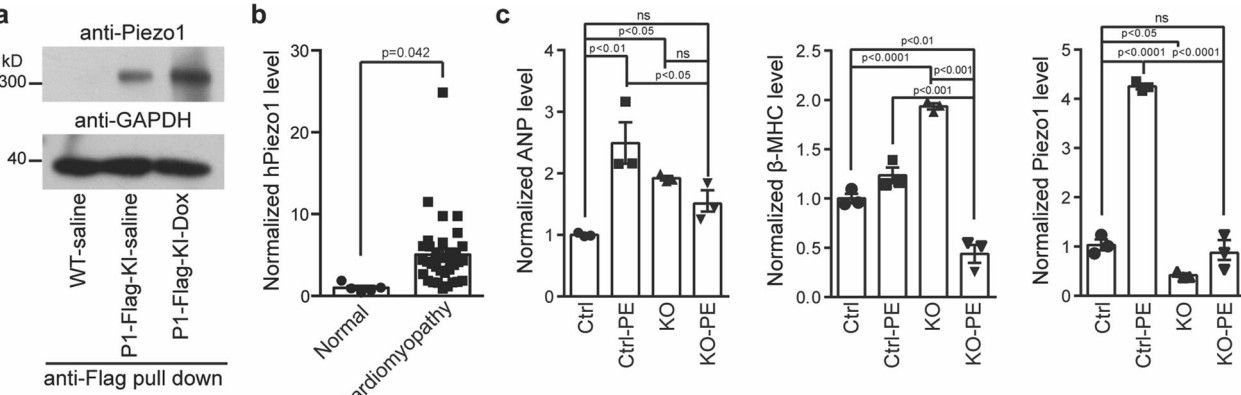

**Fig. 8 Autonomic upregulation of Piezo1 contributes to the development of cardiomyopathy. a** Western blotting of the anti-Flag-pulled-down Piezo1 proteins from adult cardiomyocytes derived from saline-treated wild-type mice or the Piezo1-Flag-KI mice treated with or without doxorubicin (Dox), which was used for inducing dilated cardiomyopathy. Similar results were obtained from 4 independent experiments. **b** RT-PCR analysis of the normalized mRNA level of human Piezo1 derived from either normal human hearts ($n = 5$) or hearts with hypertrophic obstructive cardiomyopathy ($n = 35$). Unpaired student's t-test. Values are mean ± SEM. **c** RT-PCR analysis of the normalized mRNA level of the cardiac hypertrophic genes and Piezo1 in primary neonatal ventricular myocytes derived from either control or Piezo1-KO mice treated with or without phenylephrine (PE) for 24 h, which was used for inducing cardiac hypertrophy. mRNA levels were normalized by GAPDH levels. Data from 3 independent experiments. One-way ANOVA. Values are mean ± SEM.

function and the development of cardiomyopathy of the Piezo1-KO mice (Fig. 6). On the other hand, enforcing the positive feedback loop upon overexpression of Piezo1 might initially cause increased $Ca^{2+}$ influx (Fig. 2j, k) and ROS production (Fig. 5m), resulting in enhanced spontaneous SR $Ca^{2+}$ leakage (Fig. 4d, f), deceased SR $Ca^{2+}$ store (Fig. 2i) and occurrence of arrhythmogenic $Ca^{2+}$ waves (Fig. 4g–i). Such a molecular mechanism might account for the severe heart failure and arrhythmia of the Piezo1-TG mice (Fig. 7).

A positive feedback relationship between the Piezo1 mechanosensor and the mechanical force experienced by the mechanosensitive cells and organs has been illustrated in several biological settings, in which a mechanical load leads to increased expression and activation of Piezo1. For instance, in glioma tissues, activation of Piezo1 by mechanical force upregulated extracellular matrix and stiffened the tissue, which in turn elevated Piezo1 expression to promote glioma aggression[53]. In osteoblasts, mechanical loading (e.g., mice were subjected to exercise on a treadmill) led to increased expression and activation of Piezo1, which in turn enhanced osteogenesis and bone formation. In reverse, either mechanical unloading (e.g. hind-limb suspension or microgravity treatment) or loss of the Piezo1 mechanosensor (e.g. osteoblast-specific knockout) resulted in impaired osteogenesis and bone loss[47]. A similar positive feedback mechanism might also exist between Piezo1 and the mechanical force experienced by cardiomyocytes. Indeed, an autonomic upregulation of Piezo1 in cardiomyocytes contributes to cardiomyopathy in both cellular and animal models, as well as in human patients, which might have altered mechanical stress conditions (Fig. 8a–c). Presumably, an upregulation of Piezo1 might initially serve as an adaptive response to meet the mechanical load of the heart under pathological or aging conditions, but eventually lead to cardiomyopathy due to the positive feedback mechanism on $Ca^{2+}$ and ROS signaling. Despite that the exact reason for the age-dependent phenotype remains unclear, we currently speculate that such a hypothesis might explain why only Piezo1-KO mice at 18-week old but not at 8-week old developed cardiomyopathy (Fig. 6). On the other hand, deletion of Piezo1 reversed PE-induced deterioration of cardiomyoctes (Fig. 8c). Thus, blocking Piezo1 by disrupting the positive feedback loop during pathogenesis might represent a therapeutic strategy for treating human heart diseases.

Previous studies have indicated the contribution of other $Ca^{2+}$ signaling pathways to the development of the slow force response

(SFR) of the Anrep effect[4]. For instance, on the basis of that pharmacological blocking of the $Na^+/H^+$ exchange (NHE1) reduced the SFR by ~50%, it has been hypothesized that stretch-dependent modulation of NHE1 increases intracellular $Na^+$, which in turn activates the reverse mode of the $Na^+/Ca^{2+}$ exchange (NCX) to increase intracellular $Ca^{2+}$, leading to the development of SFR[54]. However, it remains unclear whether NHE1 directly respond to mechanical stimulation or indirectly affects the bona fide cardiac mechanotransducer. Piezo1 has been shown to be inhibited by low pH[55]. One conceivable scenario is that inhibition of NHE1 results in decreased pH, which in turn inhibits stretch-induced and Piezo1-mediated $Ca^{2+}$ influx. Furthermore, as a cationic channel, Piezo1 conducts $Na^+$ as well, raising the possibility that Piezo1 might help to raise intracellular $Na^+$ and consequently activate the reverse mode of NCX to increase intracellular $Ca^{2+}$. Alternatively, Piezo1 and NHE1-NCX might function in parallel mechanotransduction pathways. Indeed, inhibition of NHE1 and NCX only suppressed ~50% of the SFR. Future studies might dissect out the relationship between Piezo1 and other molecules involved in cardiac mechanotransduction.

## Methods

**Mice.** The laboratory animal facility has been accredited by Association for Assessment and Accreditation of Laboratory Animal Care International (AAALAC) and the Institutional Animal Care and Use Committee (IACUC) of Tsinghua University approved all animal protocols used in this study.

To generate the Piezo1-Flag knock-in reporter mice (Piezo1-Flag-KI), we knocked in the Flag tag after the coding sequence corresponding to G2420 of mouse *Piezo1* using the CRISPR/Cas9 technology. T7-Cas9 PCR product and T7-sgRNA were gel purified and used as the template for in vitro transcription. Cas9 mRNA, sgRNA, and oligo were mixed and injected into the cytoplasm of one-cell fertilized eggs[56]. Two pairs of primers were designed for genotyping the genomic DNA derived from mouse tails of the Piezo1-Flag-KI breeding mice. One pair was used to distinguish between the wild-type and Flag-inserted bands, while the other pair was to verify the insertion of the Flag-tag sequence into the genome (Supplementary Table 2).

The previously described floxed Piezo1 mice (Piezo1[fl/fl])[24] (a generous gift from Dr. Ardem Patapoutian at the Scripps Research) were crossed with the MLC2v-Cre mice[39], in which the Cre recombinase coding sequence was knocked into the MLC2v genomic locus that is expressed exclusively in ventricular cardiomyocytes to obtain Piezo1[fl/fl] as littermate control (Ctrl) and Piezo1[fl/fl]/MLC2v-Cre as cardiac-specific Piezo1-knockout (KO) mice for experimental tests (Supplementary Fig. 2a). The previously generated Piezo1-TG[fl-mCherry-stop-fl] mice[40] were mated with the MLC2v-Cre mice to obtain the cardiac-specific Piezo1 transgenic mice Piezo1-TG[fl-mCherry-stop-fl]/MLC2v-Cre (Piezo1-TG) (Supplementary Fig. 2b).

Doxorubicin-induced mouse dilated cardiomyopathy (DCM) model[48] was made with intraperitoneal (i.p.) injection of doxorubicin hydrochloride (Worthington) at a dose of 3 mg/kg on 8-week-old male mice for six times over 2 weeks. One month after the last i.p. injection, mice that had a left ventricular internal diastolic diameter (LVIDd) larger than 3.8 mm or 120% of that before the injection were used as the DCM models in these experiments.

All experiments were conducted using male mice in C57/BL6 genetic background.

**Primary culture of mouse cardiomyocytes**. Adult ventricular cardiomyocytes were isolated from wild-type C57/BL6, Piezo1-Flag-KI, Piezo1-tdTomato-KI, Piezo1-KO, Piezo1-TG, and their littermate control mice. Mice were anesthetized by injection of tribromoethanol. The heart was quickly separated and perfused in a Langendorff apparatus with a zero $Ca^{2+}$ Tyrode's solution for 3 min at 37 °C, then digested with 50 µM $Ca^{2+}$ Tyrode's solution containing 0.9 mg/mL type II collagenase (Worthington, USA) and 0.05 mg/ml protease for about 20 min. All buffers contained 10 mM 2,3-butanedione monoxime (BDM) for suppressing contraction. Adult mouse cardiomyocytes were mechanically separated and gradually recovered into 1.8 mM $Ca^{2+}$ Tyrode's solution and plated on glass coverslips coated with laminin.

Neonatal mouse cardiomyocytes were isolated from neonatal mice 1–3 days after birth. After decapitation, the heart was extracted and its ventricular section was cut into pieces and sequentially digested with 0.125% Trypsin at 4 °C for 30 min and then with 0.5% collagenase II at 37 °C for 30 min. All buffers contained 20 mM BDM. Tissue fragments were filtered by a cell-strainer. The resulting cells were pre-plated for 1–2 h to allow fibroblasts and endothelial cells to attach to the culture dish. Non-adherent cardiomyocytes in the suspension were collected and transferred into a laminin-coated cell culture dish[57]. 24 h later, 100 µM of phenylephrine (PE) or vehicle was added to the cell culture, which was then maintained for an additional 24 h to induce cellular hypertrophy.

**Quantitative Real time PCR (RT-qPCR)**. Total RNA isolated from whole hearts, lungs, blood vessels, or cultured adult mouse cardiomyocytes was subjected to reverse transcription using the reverse transcription kit (Takara). Reactions were run and analyzed using a CFX96 Touch PCR instrument (Bio-Rad) with GAPDH as an endogenous internal control. All primers used were shown in Supplementary Table 2. Relative expression values for each gene were calculated using the ∆∆Ct analysis method[58].

**Immunoprecipitation and western blotting**. Cell lysates derived from isolated adult cardiomyocytes were prepared using a buffer containing 25 mM NaPIPES, 140 mM NaCl, 1% CHAPS, 0.5% phosphatidylcholine (PC), and a cocktail of protease inhibitors (Roche). For immunoprecipitation using the anti-Flag antibody, cell lysates were incubated with magnetic beads conjugated with the anti-Flag antibody at 4 °C for overnight. For immunoprecipitation using the anti-Piezo1 antibody that was custom generated by Abgent (Suzhou, China) and previously characterized[35], the protein A/G magnetic beads (Cell Signaling Technology) were incubated with either IgG or the anti-Piezo1 antibody at 4 °C for 2 h. Then the antibody-bound beads were incubated with the cell lysates at 4 °C for overnight. The immunoprecipitated beads were washed 5 times with a wash buffer containing 25 mM NaPIPES, 140 mM NaCl, 0.6% CHAPS, 0.14% PC, and a cocktail of protease inhibitors, and then boiled for 5–10 min in 1 × SDS loading buffer. For Western blot of erythrocytes from KO mice and control littermates, cell lysates were prepared using RIPA lysis buffer (Beyotime) containing a cocktail of protease inhibitors (Roche)[24]. The immunoprecipitated samples and cell lysates as loading controls were subjected to SDS-PAGE gels and electrophoresis separation. The separated proteins in the gel were transferred to PVDF membrane at 100 V for 2 h. The membrane was blocked with 5% blotting-grade blocker (Bio-rad) in TBST buffer (TBS buffer with 0.1% Tween-20) and incubated with the primary antibodies [anti-Piezo1 (1:1000), anti-GAPDH (Abcam, 1:3000) or anti-β-actin (Cell Signaling Technology, 1:3,000)] for overnight. After washing with the TBST buffer for 3 times, the membrane was incubated with the peroxidase-conjugated anti-rabbit IgG secondary antibody (CST, 1:10000) or anti-mouse IgG secondary antibody (Pierce, 1:20000) at room temperature for 1 h, followed with wash and detection using the enhanced chemiluminescence (ECL) detection kit (Pierce). All uncropped images of blots are shown in Supplementary Fig. 4.

**Measurement of body weight, heart weight, and tibia length**. Mice were weighed and then sacrificed. Hearts were rapidly removed, trimmed to remove major blood vessels, sectioned, blotted, and then weighed. After the mice were euthanized, the tibias were dissected and the tibia lengths were measured.

**Histological analysis**. Excised hearts were quickly rinsed in the PBS buffer without $Ca^{2+}$, and incubated in 4% PFA for at least 24 h at room temperature. Samples were dehydrated with ethanol, mounted in paraffin, and sectioned at 5 mm thickness. The sections were subjected to either H&E staining or Masson's trichrome staining for visualizing the tissue architecture.

**Immunocytochemistry**. Heart paraffin sections were incubated at 60 °C for 1 h and deparaffinized and rehydrated. For heat-induced antigen retrieval, the slides were boiled in a sodium citrate buffer for 10 min and then cooled down to room temperature for two times. After washing with distilled water for 5 min, the sections were permeabilized with 0.2% Triton X-100 in PBST (PBS with 0.1% Tween 20) for 10 min and blocked with 5% donkey serum in PBST for 1 h at room temperature, then incubated with the rabbit anti-Flag antibody (1:500) at 4 °C overnight, subsequently washed with PBST for 3 times. The sections were incubated with the secondary antibody, Alexa Fluor-488 donkey anti-rabbit (1:500, Invitrogen) for 1 h at room temperature. After washing with PBST for 3 times, the sections were incubated with DAPI, then dried and mounted for confocal imaging.

For adult ventricular cardiomyocytes immunostaining, cells were isolated and plated on the laminin-coated coverslips and fixed with 4% PFA for 15 min, washed 3 times with PBS, and permeabilized with 0.2% Triton X-100 for 10 min. Cells were blocked with 3% bovine serum albumin (BSA) in TBST for 1 h at room temperature, then incubated with primary antibodies and secondary antibodies for 1 h at room temperature, respectively. Anti-Flag antibody (1:500, Sigma), anti-SERCA2 (1:500, Thermofisher), anti-actinin (1:500, Sigma), anti-RyR2 (1:100, Thermofisher) were used. Secondary antibodies including Alexa Fluor 488 donkey anti-rabbit (1:500, Invitrogen), Alexa Fluor-594 donkey anti-rabbit (1:500, Invitrogen), Alexa Fluor 594 donkey-anti-mouse (1:500, Invitrogen) or Alexa Fluor 488 donkey-anti-mouse (1:500, Invitrogen) were used.

All the imaging procedures were performed on the Nikon A1 confocal microscope (Nikon Instruments) with a 100 × oil objective (N.A. = 1.49) using the 488 nm excitation wavelength and the 562 nm excitation wavelength. The images were analyzed using the Nikon NIS-Elements AR software. Co-localization of Piezo1 and SERCA2 or RyR2 either near the sarcolemma or inside the sarcoplasma of myocytes was calculated as Pearson's co-localization efficiency using the ImageJ software.

**Fura-2 single cell $Ca^{2+}$ imaging**. Cardiomyocytes grown on the laminin-coated 8-mm round glass coverslips were washed with the 1.8 mM $Ca^{2+}$ Tyrode's solution (with BDM). And then incubated with 2.5 µM Fura-2-AM (Life technologies) and 0.05% Pluronic F-127 (Life technologies) for 30 min at room temperature. The coverslip was mounted into an inverted Nikon-Tie microscopy equipped with a CoolSNAP CCD camera and Lambda XL light box (Sutter Instrument), and cells were selected for measurement of the 340/380 ratio with a 20 × objective (N.A. = 0.75) using the MetaFluor Fluorescence Ratio Imaging software (Molecular Device). 30 µM Yoda1 was used for chemical activation of Piezo1 and 10 mM caffeine for measuring SR $Ca^{2+}$ content via activating RyR2.

**Single cell $Ca^{2+}$ imaging by TIRF**. Cardiomyocytes grown on the 35 mm glass-bottom confocal dishes (Cellvis) were incubated with 2 µM Fluo-4-AM (Life technologies) for 20 min at 37 °C and an additional 10 min for de-esterification. And then scanned using a 488 nm laser in a TIRF microscopy (Nikon). Automated analysis of images was performed with Image J software installed the xySpark plugin[59].

**$Ca^{2+}$ spark and wave measurements**. Cardiomyocytes were loaded with 2 µM Fluo-4 AM for 20 min at 37 °C and an additional 10 min for de-esterification[44,60]. Cells were plated on laminin-coated 35 mm glass-bottom confocal dishes and scanned using a 488 nm laser in a confocal line-scan mode. Automated analysis of line-scan images for $Ca^{2+}$ sparks was performed using Image J software[61] and Origin9.0. For low-concentration of caffeine response, adult cardiomyocytes were scanned for 120 s and 1 mM caffeine were added into dish after 60 s record.

**Cardiomyocyte attachment and stretch**. All experiments were performed on the Nikon A1 confocal microscope (Nikon Instruments) with a 60× water objective (N.A. = 1.80) using the 488 nm excitation wavelength. The Stiff, a 25 µm diameter glass micro-rod (WPI), was coated with MyoTak™ (WPI), which is a biocompatible cellular adhesive for attaching a single cardiomyocyte to research tools without damaging the cell. One glass micro-rod was connected to a high-sensitivity force transducer (WPI), and the other to a piezo-electric length controller (WPI) driven by a variable voltage output source. The glass-rods were positioned with motorized micromanipulators (WPI) and glued to the end of a single cardiomyocyte. The attached cardiomyocytes were given either an 8% or a 15% of axil stretch of the cell length. The Fluo-4 fluorescent signals before, during, and after the stretch stimulation were recorded by line-scan mode of confocal microscopy. The data were analyzed using the ImageJ software.

**2′,7′-dichlorofluorescein diacetate (DCF) measurement of reactive oxygen species (ROS)**. Cardiomyocytes were loaded with 1 µM DCFH-DA (Beyotime) for 20 min to measure ROS[44]. Non-fluorescent DCFH could be oxidized to fluorescent DCF. Detecting the fluorescence of DCF can quantify the level of ROS in myocytes. All experiments were performed on the Nikon A1 confocal microscope (Nikon Instruments) 100× oil objective (N.A. = 1.49) using the 488 nm excitation wavelength. A total of 30 µM Yoda1 (Maybridge) was immediately added before camera shooting. For testing the blocking effect of Rac1, DCF images were taken and compared before and after application of 50 µM Rac1 inhibitor (APExBIO)

together with 30 μM Yoda1. For testing the blocking effect of NOX2, 3 μM membrane permeable gp91ds-tat (Anaspec) were pre-incubated 30 min before the application of Yoda1, and the images were taken and compared before and after the application of Yoda1. Cells were imaged at low laser intensity and the 488 nm laser was shut off during the drug delivery process to ensure that the same position was taken twice without causing artifactual signal changes of DCF.

For measuring stretch-induced ROS, cardiomyocytes were loaded with 1 μM DCF for 20 min as described above. The attached cardiomyocytes were given an 8% of axil stretch of the cell length and recorded DCF fluorescence intensity. DCF loaded cells were imaged using confocal line-scanning microscopy. As DCF can produce artifactual signal amplification upon continuous light exposure, cardiomyocytes were imaged at low laser intensity. Meanwhile, the attached cells were first recorded for the baseline fluorescence without stretch, then followed by stretch-induced fluorescence change. The baseline fluorescence was subtracted from the stretch-induced fluorescence change to avoid artifactual fluorescence change caused by illumination of DCF itself. Confocal line-scans were taken from the interior of the myocytes in the stretched region of the cells.

**Echocardiographic analysis**. Mice were anesthetized and the cardiac function was evaluated on conscious mice using a Vevo 2100 Ultrasound system. The images of long-axis view of left ventricle (LV), short-axis view of LV, and apical four-chamber view were obtained (Supplementary Table 1a–f).

**Electrocardiogram (ECG) recoding**. Mice were lightly anesthetized with isoflurane vapor (0.5–1%) and 95% $O_2$. Anesthetized mice were placed on a pad, and subcutaneous needle electrodes were inserted into the right upper limb and hind limbs for continuous ECG recordings lasting at least 5 min after the heart rate became stabilized.

**Human heart samples**. The study was approved by the Ethics Committee of Fuwai Hospital and adhered to the Declaration of Helsinki. The study of the human heart samples was approved by the ethics committee of the institutional review board at Fuwai Hospital in Beijing. All experiments were complied with all relevant ethical regulations. All the patients who participated in the study provided written informed consent prior to sample collection and all other procedures and we have obtained consent to publish the information. The human sample information is shown in the Supplementary Data 1.

**Data analysis**. All data are shown as mean ± standard error of mean (SEM). Statistical significance was evaluated using either unpaired or paired Student's t-test for analyzing two groups of samples, one-way or two-way ANOVA for analyzing multiple groups of samples.

## Data availability

All relevant data are available from the corresponding author upon reasonable request. Source data are provided with this paper.

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

## Acknowledgements

We thank Dr. Ardem Patapoutian for sharing the Piezo1$^{fl/fl}$ mice; Dr. Wayne Chen for critical discussion; Dr. Zai Chang and other stuff at the animal facility center of Tsinghua University for maintaining mice; Yalan Chen at the Imaging Core Facility, Technology Center for Protein Sciences at Tsinghua University for assistance with Nikon A1 confocal microscopy; Dr. Qing Xu at the Core Facilities Center at the Capital Medical University for assisting in echocardiography and hemodynamic experiments. This work was supported by grants from the the National Key R&D Program of China (2016YFA0500402) and the National Natural Science Foundation of China (31630090 and 31825014) to B.X.

## Author contributions

F.J. carried out the bulk of the experiments and analyzed data; K.W. and M.Z. helped the initial characterization of the Piezo1-TG mice; H.C. and S.W. provided technique help; K.Y. and Z.Z. provided the human heart samples; B.X. conceived and directed the study, analyzed data, and wrote the manuscript with help from all other authors.

## Competing interests

The authors declare no competing interests.
