## [Peer Review File · Nature Communications]

REVIEWER COMMENTS

Reviewer #1 (Remarks to the Author):

Dear authors,

This study examined the expression and role of Piezo1 in cardiomyocytes and heart function using cardiac-specific knockout and overexpression model mice. Firstly, the authors showed that Piezo1 in cardiomyocytes converted mechanical force into Ca²⁺ and reactive oxygen species (ROS) signaling. Secondly, they showed that either deletion or overexpression of Piezo1 resulted in dysregulated Ca²⁺ and ROS signaling and cardiac dysfunction. From these results, the authors would like to propose that Piezo1-mediated mechano-chemo transduction process might safeguard a homeostatic functional state of the heart. Finally, the authors showed that Piezo1 was increased in the cardiac tissue of human patients with heart failure, and that the deletion of Piezo1 from mice hearts showed resistance to doxorubicin-induced deterioration of contractile function. Taken together, the authors would like to suggest that blocking Piezo1 might represent a novel therapeutic strategy for treating human heart disease. Mechano-sensitive clues are important for organogenesis, physiological and pathological tissue remodeling, and electro-mechanical responses. So the research topic presented here has importance. However, Piezo1 protein expression level in the cardiac tissue is extremely low. In addition, Piezo1 activity or effects in the current study was estimated using a high concentration of Yoda1-treatment for a long exposure. Therefore, as a reviewer I am concerned about whether Piezo1 protein has some physiological role in the heart. In addition, other major questions about the trigger event and molecular mechanism induced by elimination or overexpression of Piezo1 in myocytes of mice needs to be addressed. Therefore, a slight trend towards 'causal' interpretation of what may simply be correlations are weaker points of the manuscript. The conclusion suggesting that Piezo1 serves as the long-sought-after cardiac mechanosensor, which is involved in the Frank-Starling law and Anrep effect is greatly over-interpreted.

1. Previous studies and this study have shown Piezo1 expression level in cardiac tissue is extremely low. In this study, authors can't detect Piezo1 proteins from heart homogenates without immunoprecipitation using the anti-Piezo1 antibody. In addition, to detect the activity of Piezo1, the authors used a high concentration of Yoda1 in Fig. 1h (30 μ M). It's also strange that Yoda1-induced Ca²⁺ increase is very slow (Fig. 1h). For example, in the case of HEK cell expressing Piezo1 cDNA, IC50 is about 1.3 μ M. The intracellular Ca²⁺ quickly increases after Yoda1-treatments in various cells. In Fig. 1h of this study, the time to peak is about 30s after Yoda1-treatment. Did myocyte contraction occur during Yoda1-treatments? If Piezo1 modified E-C coupling and contraction of myocytes, Yoda1-induced Ca²⁺ increase should show more of a quick response. Although the expression and activity declined in cKO myocytes compared with control myocytes, the levels are extremely low in both myocytes. Therefore, as a reviewer I am wondering if this makes sense. To

address this issue, the authors have to reduce the concentration of Yoda1 for Ca²⁺ imaging experiments (about several μM). In addition, the authors need to analyze the significance of the effects in consideration of caffeine-induced Ca²⁺ release (SR contents) of the myocytes. In Fig. 2m, authors showed that the caffeine response in cKO myocytes was about half of control myocytes. From this data, as a reviewer I am also concerned about the remodeling of the cKO myocytes.

2. In Figure 2, the authors show that stretch-induced Ca²⁺ spark was not observed in cKO myocytes. It is strange that the Ca²⁺ spark rate in baseline (pre-stretch condition) in cKO myocytes is different in the control (Fig. 2b). In addition, the spontaneous Ca²⁺ spark events and its amplitude in cKO myocytes are revealed to be significantly less than in control myocytes (Fig. 2d and e). This probably reflects the reduction of SR Ca²⁺ contents in cKO myocytes (Fig. 2m). Thus, as a reviewer I am concerned whether the lack of stretch-induced Ca²⁺ spark in cKO myocytes is dependent on the reduction of stretch-sensitivity induced by elimination of Piezo1, or some sort of functional deficiency of myocytes induced by deletion of Piezo1 protein from early stage of embryonic mice. Because this cardiac-specific Piezo1-deficient mice was generated using MLC2v (myosin light chain 2v)-cre mice. The expression of MLC2v starts from the early stage of embryonic mice (E.7.5 days).

3. In Figure 2, the authors show the stretch-induced Ca²⁺ spark in control myocytes. Does Yoda1 (~ several μM) also enhance the Ca²⁺ spark in control myocytes?

4. A previous study (Iribe G et. al., 2009) has reported that 2 $\mu\text{mol/L}$ GsMTx-4 (blocker of stretch-activated ion channels) had no effect on the stretch-induced acute increase in Ca²⁺ spark rate. Because GsMTx-4 is one of the inhibitors of Piezo1, this current study contradicts the previous study.

5. Piezo1-TG mice were also generated by using MLC2v-cre mice. The Piezo1 protein was expressed from the early stage of embryonic mice (E.7.5 days). Therefore, myocyte growth and maturation has affected the overexpression of Piezo1 protein in the myocytes isolated from Piezo1-TG mice. Then, the authors can't say that the disordered resting Ca²⁺ homeostasis in the myocytes depend on elimination of Piezo1.

6. In the current study, the authors analyzed the effects of Yoda-1 treatment with high concentration and long exposure (Fig. 1 and 3).

7. Piezo1-deficient mice showed no abnormality in cardiac function at 8 weeks, suggesting that Piezo1 is not absolutely necessary for the cardiac normal function in this stage. The mild dilation of left ventricle observed in cKO hearts at 18 weeks is due to the pathological remodeling of cardiomyocytes with mild dilation of left ventricle (Fig. 4d). Thus, this does not suggest a

compromised Frank-Starling response in cKO hearts. Lack of Piezo1 from the embryonic stage is probably likely to affect the various properties in homeostasis to the heart aging in these myocytes.

8. Cardiac-specific overexpression of Piezo1 induces severe heart failure and arrhythmia (Fig. 5). Usually, whether the protein is expressed or not in myocytes, the overexpression of Ca²⁺ transporter leads to heart failure. This TG mice also overexpress Piezo1 from the embryonic stage. The authors need to address the trigger event and molecular mechanism induced by the overexpression of Piezo1 in myocytes of mice.

9. The authors showed the upregulation of Piezo1 protein in doxorubicin-induced heart failure mice hearts and human heart sample. As you know, Piezo1 protein is highly express in the vessel and blood. Upregulation of Piezo1 is possibly the result from the remodeling of coronary artery in pathology. The authors should compare Piezo1 mRNA normalized by internal control (ex, GAPDH) in neonatal myocytes with or without PE-treatment.

10. This report suggested deletion of Piezo1 might prevent doxorubicin (DOX)-induced heart failure phenotype in the cKO mice. Since the loss of contractile function happened upon Piezo1-deletion, it is very difficult to evaluate its response to DOX. The overall phenotype was a result of defects in contractility combined with much slower (weeks) and less severe defects induced by DOX. Therefore, the conclusion about the role of Piezo1 protein in DOX-induced heart failure was not a valid one. This part of the study was not well designed and could not offer a clear insight to the Piezo1 function in DOX-induced heart failure. In this study, this is a distraction and should be removed.

Reviewer #2 (Remarks to the Author):

This study investigates a role for the mechanosensitive cation channel Piezo1 mechano-chemo transduction in normal and diseased hearts. The authors utilized several mouse models, including the previously established Piezo1-tdTomato knock-in model, wild-type C57/BL6 mice, and several new mouse models. New models include a Piezo-Flag knock-in reporter mouse, a cardiac-specific Piezo1 knockout mouse, and a cardiac-specific Piezo1 overexpressing mouse. Protein expression and colocalization were assessed using immunocytochemistry, immunoprecipitation, western blotting, and PCR. Heart structure, function, and remodeling were determined via histology, measurement of

body weight and heart weight, echocardiography, and electrocardiography. Calcium signaling was assessed via Fura-2 single cell calcium imaging, and ROS signaling was assessed via 2',7'-dichlorofluorescein diacetate assays. The authors report that Piezo1 is expressed in the sarcolemma of adult cardiomyocytes, and that Piezo1 plays a role in mediating both stretch-induced and homeostatic calcium and ROS signaling. Additionally, they demonstrated that cardiac-specific knockout of Piezo1 impairs heart function and that cardiac-specific overexpression of Piezo1 may induce severe heart failure and arrhythmias. Finally, they report that autonomic upregulation of Piezo1 contributes to the development of cardiomyopathy. A clear strength of this paper is the use of novel mouse models. The Piezo-Flag knock-in reporter mouse, cardiac-specific Piezo1 transgenic mouse, and cardiac-specific Piezo1 knockout mouse are all excellent tools for exploring the questions asked in this paper, as well as for asking future questions. Importantly, a range of techniques are used to assess cardiac function, including echocardiography, ECG, heart weight, and fibrosis analysis, was also a strength. They used appropriate controls, and overall, their methods seemed to be appropriate, especially their assessments of calcium signaling and their induction of cell stretching. That said, there are some concerns with the approach and data interpretation that need to be addressed.

Specific comments:

In presenting their arrhythmia data from the transgenic mice, the authors only present representative ECG traces from 4-week-old, 8-week-old, and 18-week-old mice (Fig. 5h). It would be preferable if they quantified the occurrence of arrhythmias in some way (even just incidence of VT, PVCs, or alternans), because with just the representative traces, it is difficult to appreciate the nature or extent of arrhythmia in overexpressing mouse.

There is some inconsistency in reporting of the data. For confirming protein expression in their different mouse models, they show mRNA levels and western blots for certain mice and immunostaining for others (Fig. 1a-b, e-g). To be rigorous, they should provide both sets of data (mRNA levels and western blots as well as immunostaining) for each mouse model in which they're examining expression. Additionally, DCF fluorescence data are reported for the control cardiomyocytes before and after Yoda1 treatment in the absence of 1.8 mM calcium, but they do not report this data for KO cardiomyocytes (Fig 3a-f). For some calcium signaling experiments, they report number of cells examined, but for others they report number of animals and number of cover slips. For consistency, they should report the number of cells in addition to the number of animals and cover slips, where applicable (Fig. 1h-j, Fig. 2c, e-f, h-l, l-m).

The authors also are not clear what sex of mice they utilized, and it is important to utilize both male and female mice to avoid confounding their data by sex effects. Also, what are background of mice used in doxorubicin study?

Zoomed in images don't seem to align with yellow boxed regions in Figure 1a.

Hearts of KO appear larger by histology with increased chamber dimensions by echo but HW actually decreased in Piezo1-KO – what is the explanation?

The authors come to the conclusion that “the mechanosensitive Piezo1 channel serves as the long-sought-after cardiac mechanotransducer that directly converts mechanical stretch of cardiomyocytes into Ca²⁺ and ROS signaling.” This seems like a bit of a lofty statement – although they demonstrate that it plays a role in translating mechanical stretch into calcium and ROS signaling, they are going a bit far to say that it is the “long-sought-after cardiac mechanotransducer.”

Reviewer #3 (Remarks to the Author):

The authors report a novel role for stretch-activated ion channel Piezo1 in the slow force response (SFR) of Ca²⁺ and ROS regulation in cardiomyocytes as an in vitro equivalent to the Anrep effect. They also demonstrate cardiomyopathies associated with KO and overexpression of Piezo1 in mice.

This is a comprehensive study that represents a novel and important contribution that will be of interest to the field. In particular, the demonstration of a function effect of Piezo1 towards the Anrep effects and a potential involvement in cardiac pathologies are novel. However, some claims are not fully supported by the data and methodological and statistical clarification is needed. In particular:

1. The authors do not account for previously-demonstrated mechanisms in the SFR, such as the crosstalk between Na/H and Na/Ca exchange (such as Luers et al. (2005) *Cardiovasc Res* (10.1016/j.cardiores.2005.07.001); and the additional citing literature that follows). The current work should be discussed in this context.

2. The authors state “Here we have established that the mechanosensitive Piezo1 channel serves as the long-sought-after cardiac mechanotransducer that directly converts mechanical stretch of cardiomyocytes into Ca²⁺ and ROS signaling”, but do not adequately demonstrate a contribution to the Frank-Starling (or rapid) stretch response and do not acknowledge the other demonstrated mechanisms of Ca stretch response. Claims should be tempered and qualified.

3. “The observation that a stretched diastolic left ventricle of the KO hearts ($81.8 \pm 8.7 \mu\text{l}$) relative to that of the control hearts ($61.2 \pm 3.4 \mu\text{l}$) led to a comparable stroke volume suggests a compromised Frank-Starling response in the KO heart. The KO hearts had a concomitant decreased ejection fraction and fractional shortening”. This is a description of DCM phenotype. Without testing each condition’s response to an acute preload, a compromised F-S effect is not a logically-sound argument. Even then, the changes to anatomy based on the DCM-like phenotype provide a confounding factor to these measurements.

4. The stretch experiments were performed with 8% longitudinal strain, which is well below the 15-20% that would represent a healthy beat. This may underscore the predominance of Piezo1 in the SFR demonstrated in this paper and should be discussed.

5. The composition of buffers used in experiments must be defined, given the relationships between ionic environment and the function of ionic channels. Additionally, confirm “50 mM Ca²⁺ Tyrode’s solution containing 0.9 mg/mL type II collagenase”. Was this supposed to be 50 mg/L? (approx. 33% of physiological Ca²⁺)? Please provide composition of Tyrode’s solution used as most dissociations use Ca-free, Mg-substituted media with collagenase; if Ca was indeed at 50 mM, please provide solubilization steps and measures to balance osmolarity.

6. Piezo1 knockout had significantly lower baseline levels of Ca²⁺ sparks (Fig. 2A,B), confounding the interpretation of the lack of stretch-response and lack of stretch-induced ROS production. Does stretch induce Ca²⁺ sparks in TG cells subjected to stretch? Are Ca²⁺ sparks inducible in the KO cells by non-mechanical stimuli?

7. The proposed pathway in 3i is not fully supported by the data presented. In particular, do Yoda1 or stretch activate Nox2 or ROS and to similar degrees?

8. The caption in Fig 1J implies that multiple coverslips were used from a single mouse and treated as independent samples in the statistical analysis. Please clarify and justify.

9. The normalization employed in Fig 3j and 3k yields no variance in the “before” samples and thus inappropriate t-tests. Please clarify the normalization procedure in the methods, or reanalyze data for j and k as was done previously in Fig 3b,c,d,f,g.

10. In Extended Data Fig 4: These analyses represent multiple comparisons. They should be presented in a single panel, and should be analyzed by 2-way ANOVA followed by a post-hoc test of choice. This would also carry a benefit in assessing main effects and the potential for interaction.

11. Patient characteristics for human heart tissue are missing. What types of CM are represented in the disease group?

Reviewer #4 (Remarks to the Author):

The beating heart possesses intrinsic ability to adapt cardiac output to changes in mechanical load. The century-old Frank-Starling law and Anrep effect have documented that stretching heart during diastolic filling increases its contractile force. However, the molecular mechanotransduction mechanism and its impact on cardiac health and disease remain elusive. Here we show that the mechanically activated Piezo1 channel converts mechanical stretch of cardiomyocytes into Ca²⁺ and reactive oxygen species (ROS) signaling, which critically determines the mechanical activity of the heart. Either cardiac-specific knockout or overexpression of Piezo1 in mice results in defective Ca²⁺ and ROS signaling and the development of cardiomyopathy, demonstrating a homeostatic role of Piezo1. Piezo1 is pathologically upregulated in both mouse and human diseased hearts, and its deletion renders mice with dilated cardiomyopathy resistant to deterioration of cardiac function. Thus, Piezo1 serves as the long-sought-after cardiac mechanotransducer for initiating mechano-chemo transduction and consequently maintaining normal heart function, and might represent a novel therapeutic target for treating human heart diseases. Also this work is interesting, a number of concerns remain.

1. The degree (extend) of stretch makes any difference? A couple of more points may be worthy trying.

2. What is the rationale of using 18 months? Any data for survival for the overexpression mice?

3. What may be the precise mechanism for Piezo1-induced calcium regulation?

4. Any effects on mitochondrial calcium – any ultrastructural changes?

Reponses to reviewers' comments

Reviewer #1 (Remarks to the Author):

Dear authors,

This study examined the expression and role of Piezo1 in cardiomyocytes and heart function using cardiac-specific knockout and overexpression model mice. Firstly, the authors showed that Piezo1 in cardiomyocytes converted mechanical force into Ca²⁺ and reactive oxygen species (ROS) signaling. Secondly, they showed that either deletion or overexpression of Piezo1 resulted in dysregulated Ca²⁺ and ROS signaling and cardiac dysfunction. From these results, the authors would like to propose that Piezo1-mediated mechano-chemo transduction process might safeguard a homeostatic functional state of the heart. Finally, the authors showed that Piezo1 was increased in the cardiac tissue of human patients with heart failure, and that the deletion of Piezo1 from mice hearts showed resistance to doxorubicin-induced deterioration of contractile function. Taken together, the authors would like to suggest that blocking Piezo1 might represent a novel therapeutic strategy for treating human heart disease. Mechano-sensitive clues are important for organogenesis, physiological and pathological tissue remodeling, and electro-mechanical responses. So the research topic presented here has importance. However, Piezo1 protein expression level in the cardiac tissue is extremely low. In addition, Piezo1 activity or effects in the current study was estimated using a high concentration of Yoda1-treatment for a long exposure. Therefore, as a reviewer I am concerned about whether Piezo1 protein has some physiological role in the heart. In addition, other major questions about the trigger event and molecular mechanism induced by elimination or overexpression of Piezo1 in myocytes of mice needs to be addressed. Therefore, a slight trend towards 'causal' interpretation of what may simply be correlations are weaker points of the manuscript. The conclusion suggesting that Piezo1 serves as the long-sought-after cardiac mechanosensor, which is involved in the Frank-Satrling law and Anrep effect is greatly over-interpreted.

Response: We thank the reviewer for the thorough and critical review of the manuscript. After addressing the reviewer's comment either via new experiments or clarification and discussion, we believe that the revised version of the manuscript has been significantly improved.

1.Previous studies and this study have shown Piezo1 expression level in cardiac tissue is extremely low. In this study, authors can't detect Piezo1 proteins from heart homogenates without immunoprecipitation using the anti-Piezo1 antibody. In addition, to detect the activity of Piezo1, the authors used a high concentration of Yoda1 in Fig. 1h (30 μ M). It's also strange that Yoda1-induced Ca²⁺ increase is very slow (Fig. 1h). For example, in the case of HEK cell expressing Piezo1 cDNA, IC₅₀ is about 1.3 μ M. The intracellular Ca²⁺ quickly increases after Yoda1-treatments in various cells. In Fig. 1h of this study,

the time to peak is about 30s after Yoda1-treatment. Did myocyte contraction occur during Yoda1-treatments? If Piezo1 modified E-C coupling and contraction of myocytes, Yoda1-induced Ca^{2+} increase should show more of a quick response. Although the expression and activity declined in cKO myocytes compared with control myocytes, the levels are extremely low in both myocytes. Therefore, as a reviewer I am wondering if this makes sense. To address this issue, the authors have to reduce the concentration of Yoda1 for Ca^{2+} imaging experiments (about several μM). In addition, the authors need to analyze the significance of the effects in consideration of caffeine-induced Ca^{2+} release (SR contents) of the myocytes. In Fig. 2m, authors showed that the caffeine response in cKO myocytes was about half of control myocytes. From this data, as a reviewer I am also concerned about the remodeling of the cKO myocytes.

Response: Although previous studies have reported relatively low level of Piezo1 mRNA expression in heart tissues and primarily cultured cardiomyocytes¹⁻³, the abundance and localization of endogenous Piezo1 proteins in cardiomyocytes have not been characterized. Using both the Piezo1-Flag-KI and Piezo1-tdTomato-KI mice, we have clearly detected the expression and localization of endogenous Piezo1 proteins in sarcolemma of ventricular cardiomyocytes either by immunostaining of heart tissues or primarily cultured cardiomyocytes (revised Fig. 1a-d). Interestingly, Piezo1 in cardiomyocytes is mainly localized in the sarcolemma including T tubules, where it co-localizes with SERCA2 residing in the sarcoplasm (revised Fig. 1c). Thus, the expression and localization of Piezo1 clearly supports its role as a mechanotransduction channel for mediating Ca^{2+} influx in cardiomyocytes.

Yoda1 is a chemical activator of Piezo1 with an apparent EC_{50} (the concentration for causing half maximal activation) of 17 μM and a maximal water solubility of 30 μM originally reported by Patapoutian laboratory⁴. In our own hands, we have found similar activation properties of Yoda1⁵. Beech laboratory has reported an EC_{50} of 2.5 μM and 0.23 μM for Piezo1 stably expressed in HEK293T-Rex cells and HUVECs, respectively. The variable effect of Yoda1 might be dependent on cell types and its low water solubility. To assay the functionality of Piezo1 in cardiomyocytes, we thus have decided to use 30 μM Yoda1 for maximal activation of endogenously expressed Piezo1 in cardiomyocytes. For comparison, we used 10 mM caffeine to activate RyR2 for measuring the total SR Ca^{2+} content, which is a key determinant of cardiac Ca^{2+} signaling and excitation-contraction coupling. For Ca^{2+} imaging of cultured cardiomyocytes, we have included 10 mM 2,3-butanedione monoxime (BDM) in the buffer to inhibit their contraction. The following results suggest that Piezo1 mediates relatively robust Yoda1-induced Ca^{2+} influx in control cardiomyocytes, which was nearly abolished in the Piezo1-KO cells.

1) A sustained Yoda1-induced Ca^{2+} response was detected in cardiomyocytes derived from the littermate control mice in the presence of 1.8 mM extracellular Ca^{2+} (revised Fig. 2c). Removing the extracellular Ca^{2+} abolished the response

(revised Fig. 2e), suggesting Yoda1-induced Ca^{2+} influx.

2) The amplitude of the Ca^{2+} increase reached about 94% of the caffeine response (revised Fig. 2c, d, f, g), suggesting relatively robust Yoda1-induced Ca^{2+} increase in cardiomyocytes.

3) The Yoda1 response was nearly completely abolished in cardiomyocytes derived from the Piezo1-KO mice (revised Fig. 2c, d), demonstrating that the Yoda1 response was specifically mediated by Piezo1.

4) Cardiomyocytes derived from the Piezo1-TG mice had significantly larger Yoda1-induced Ca^{2+} response than that detected in the littermate control cells (revised Fig. 2j, k).

5) We have carried out new experiments by using total internal reflection fluorescent microscopy to image Yoda1-induced localized Ca^{2+} influx near the sarcolemma. We detected rapid and robust Yoda1-induced Ca^{2+} events in control cardiomyocytes, which were nearly abolished in the KO cells (new Fig. 2h, i).

6) Compared to caffeine-induced Ca^{2+} response (revised Fig. 2f), the Yoda1-induced response in both control and Piezo1-TG cells appeared to be slower, but more sustained (revised Fig. 2c, j). These data suggest that the relatively slow global increase of cytosolic Ca^{2+} evoked by Yoda1 was not caused by the expression level of Piezo1.

7) We noticed that the onset of Yoda1-induced localized Ca^{2+} influx (new Fig. 2h) appeared to be much faster than Yoda1-induced global cytosolic Ca^{2+} increase assayed by single-cell Ca^{2+} imaging (revised Fig. 2c, j). Thus, we reasoned that the Yoda1-induced global cytosolic Ca^{2+} increase might result from a summation of unsynchronized Ca^{2+} influx mediated by Piezo1 distributed in the large sarcolemma area including T-tubules (revised Fig. 1), resulting in the relatively slow onset of the Yoda1 response observed in single-cell Ca^{2+} imaging.

8) Both an 8% and 15% stretch of the control cardiomyocytes evoked an apparent and rapid increase in Ca^{2+} sparks, which was not observed in the Piezo1-KO cardiomyocytes (revised Fig. 3a, b). These data suggest that Piezo1 can mediate rapid response to mechanical stimulation in cardiomyocytes.

Based on the above expression and functional assays, we believe that Piezo1 is optimally expressed in cardiomyocytes to mediate relatively robust Ca^{2+} influx in cardiomyocytes. We have included the above results and description in the revised Fig. 1 and Fig. 2 and in the Result section of “Expression and localization of Piezo1 proteins in cardiomyocytes” and “Piezo1 mediates Yoda1 induced Ca^{2+} responses in cardiomyocytes” in the revised manuscript. In the second paragraph of the Discussion section, we have also included the discussion that “Such a positive feedback signaling transduction mechanism might allow a relatively low abundant but well sarcolemma-distributed Piezo1 to optimally respond to the drastic and repeated mechanical stress generated by the beating heart”.

2. In Figure 2, the authors show that stretch-induced Ca^{2+} spark was not observed in cKO myocytes. It is strange that the Ca^{2+} spark rate in baseline (pre-stretch condition) in cKO myocytes is different in the control (Fig. 2b). In addition, the spontaneous Ca^{2+} spark events and its amplitude in cKO myocytes are revealed to be significantly less than in control myocytes (Fig. 2d and e). This probably reflects the reduction of SR Ca^{2+} contents in cKO myocytes (Fig. 2m). Thus, as a reviewer I am concerned whether the lack of stretch-induced Ca^{2+} spark in cKO myocytes is dependent on the reduction of stretch-sensitivity induced by elimination of Piezo1, or some sort of functional deficiency of myocytes induced by deletion of Piezo1 protein from early stage of embryonic mice. Because this cardiac-specific Piezo1-deficient mice was generated using MLC2v (myosin light chain 2v)-cre mice. The expression of MLC2v starts from the early stage of embryonic mice (E.7.5 days).

Response: We thank the reviewer for this constructive comment and we agree with the reviewer that the lower spontaneous Ca^{2+} spark events might reflect the lower SR Ca^{2+} store of the cKO myocytes. To exclude the possibility that the lack of stretch-induced Ca^{2+} spark in the KO cells might be due to their decreased SR Ca^{2+} content, we have carried out new experiments to use 1 mM caffeine to trigger Ca^{2+} sparks (new Fig. 3f, g). Despite relatively lower Ca^{2+} spark events in the KO cardiomyocytes due to their reduced SR Ca^{2+} content, the caffeine-induced fold change of Ca^{2+} sparks were similar between the control and KO cells (new Fig. 3g). These data suggest that the Piezo1-KO cardiomyocytes specifically lost stretch-induced but retained caffeine-induced Ca^{2+} sparks. Taken together, these data demonstrate that Piezo1 mediates stretch-induced Ca^{2+} sparks in cardiomyocytes. We have included these new data in Fig. 3f, g and described in the section of “Piezo1 mediates stretch-induced Ca^{2+} sparks in cardiomyocytes” in the revised manuscript.

When examined at 8-week old, the cardiac-specific Piezo1-KO mice did not display obvious defects in heart and body weights, heart morphology and pump function (Supplementary Fig. 3a-e and Supplementary Table 1a). Echocardiographic analysis revealed no apparent structural defects in the left ventricles of the KO mice (Supplementary Movie1). Thus, the Piezo1-KO mice at 8-week old showed overall normal heart structure and function, suggesting that the MLC2v-Cre dependent deletion of Piezo1 might not cause developmental defects of the heart during the embryonic developmental stage.

3. In Figure 2, the authors show the stretch-induced Ca^{2+} spark in control myocytes. Does Yoda1 (~ several μM) also enhance the Ca^{2+} spark in control myocytes?

Response: We thank the reviewer for this constructive comment. Following the reviewer's suggest, we have carried out new experiments by using total internal reflection fluorescent microscopy to image localized Ca^{2+} influx near the plasma membrane. We detected rapid and robust Yoda1-induced Ca^{2+} events in control

cardiomyocytes, which were nearly abolished in the KO cells (new Fig. 2h, i). We have included the new data in Fig. 2h, i and description of the data in the Result section of “Piezo1 mediates Yoda1 induced Ca²⁺ responses in cardiomyocytes” in the revised manuscript.

4. A previous study (Iribe G et. al., 2009) has reported that 2 μmol/L GsMTx-4 (blocker of stretch-activated ion channels) had no effect on the stretch-induced acute increase in Ca²⁺ spark rate. Because GsMTx-4 is one of the inhibitors of Piezo1, this current study contradicts the previous study.

Response: Previous studies have suggested that stretch-induced Ca²⁺ sparks are mediated by a mechano-chemo signaling pathway termed X-ROS signaling, which involves stretch-activation of NOX2 via Rac1-dependent activation of microtubules, leading to the production of ROS that in turn modulates the activity of RyR2⁶. However, whether mechanosensitive ion channels are involved in the X-ROS signaling has remained unclear. Using GsMTx4 as a blocker for mechanosensitive ion channels, one study has shown that GsMTx4 had no effect on the stretch-induced acute increase in Ca²⁺ spark rate⁷, while another study has observed blocking effect⁸. We therefore asked whether Piezo1 might be involved in this X-ROS signaling pathway by using the Piezo1-KO mouse model. Remarkably, both Yoda1 and stretch-induced ROS production were completely abolished in the Piezo1-KO cardiomyocytes (revised Fig. 5a-h). Thus, we believe that our data provide compelling genetic evidence that Piezo1 mediates both Yoda1- and stretch-induced production of ROS in cardiomyocytes. The reason for the previous inconsistent results regarding the use of GsMTx4 is unclear. Despite that GsMTx4 is able to block heterologously expressed Piezo1, its effect on endogenously expressed Piezo1 in cardiomyocytes might be variable. We have included these description in the Result section of “Piezo1 mediates stretch-induced and homeostatic ROS signaling” in the revised manuscript.

5. Piezo1-TG mice were also generated by using MLC2v-cre mice. The Piezo1 protein was expressed from the early stage of embryonic mice (E.7.5 days). Therefore, myocyte growth and maturation has affected the overexpression of Piezo1 protein in the myocytes isolated from Piezo1-TG mice. Then, the authors can't say that the disordered resting Ca²⁺ homeostasis in the myocytes depend on elimination of Piezo1.

Response: Crossing the MLC2v-Cre mice with the Piezo1^{fl/fl} mice to generate the Piezo1 conditional KO mice, we have confirmed the successful and specific deletion of Piezo1 in the heart, but not in lung, blood vessel and red blood cells (Fig. 2a and Supplementary Fig. 2c), suggesting specific deletion of Piezo1 in cardiomyocytes. Thus, we expected specific overexpression of Piezo1 in the Piezo1-TG mice using the MLC2v-cre mice. Despite their enlarged hearts of the TG hearts, there was no significant difference in tibia length between the TG and control mice (new Fig. 7e), indicating that the overall growth and size

of the TG mice was normal. Furthermore, despite their severely impaired heart pump function, echocardiographic analysis revealed no structural defects in the left ventricle of TG mice (Supplementary Movie2), indicating that the defective pump function of the TG mice might not be due to developmental defects.

6. In the current study, the authors analyzed the effects of Yoda-1 treatment with high concentration and long exposure (Fig. 1 and 3).

Response: Please refer our response to the above comment 1.

7. Piezo1-deficient mice showed no abnormality in cardiac function at 8 weeks, suggesting that Piezo1 is not absolutely necessary for the cardiac normal function in this stage. The mild dilation of left ventricle observed in cKO hearts at 18 weeks is due to the pathological remodeling of cardiomyocytes with mild dilation of left ventricle (Fig. 4d). Thus, this does not suggest a compromised Frank-Starling response in cKO hearts. Lack of Piezo1 from the embryonic stage is probably likely to affect the various properties in homeostasis to the heart aging in these myocytes.

Response: Indeed, only 18-week old, but not 8-week-old cKO mice showed a significantly increased end-diastolic and end-systolic internal diameter and volume of left ventricle (revised Fig. 6 i-k and Supplementary Table 1b), a concomitant decreased ejection fraction and fractional shortening (revised Fig. 6l, m and Supplementary Table 1b), demonstrating impaired heart pump function. Given that the Piezo1-cKO mice at 8-week old showed overall normal heart structure and function, we hope the reviewer would agree with us that the MLC2v-Cre dependent deletion of Piezo1 might not cause developmental defects of the heart during the embryonic developmental stage. While the reason for this age-dependent phenotype remains unclear, we speculated in the Discussion section that “Presumably, an upregulation of Piezo1 might initially serve as an adaptive response to meet the mechanical load of the heart under pathological or aging conditions, but eventually lead to cardiomyopathy due to the positive feedback mechanism on Ca^{2+} and ROS signaling. Such a hypothesis might explain why only older Piezo1-KO mice developed cardiomyopathy (revised Fig. 6).

8. Cardiac-specific overexpression of Piezo1 induces severe heart failure and arrhythmia (Fig. 5). Usually, whether the protein is expressed or not in myocytes, the overexpression of Ca^{2+} transporter leads to heart failure. This TG mice also overexpress Piezo1 from the embryonic stage. The authors need to address the trigger event and molecular mechanism induced by the overexpression of Piezo1 in myocytes of mice.

Response: We have used the Piezo1-TG mice as a gain-of-function approach to comprehensively address the role of Piezo1 in cardiomyocytes and heart function and disease. Given that both mouse models and human patients of cardiomyopathy showed increased expression of Piezo1, we believe that

studying the Piezo1-TG mice might provide important insights into the role of Piezo1 in the pathogenesis of cardiomyopathy. Mechanistically, the observation that Piezo1 controls both stretch-induced and the homeostasis of both Ca^{2+} and ROS might suggest a Piezo1-mediated positive feedback loop between these two signaling molecules critical for cardiac function. Piezo1-dependent Ca^{2+} influx regulates the production and homeostasis of ROS via the Rac1-NOX2 signaling pathway, which might in turn modulate Ca^{2+} release via acting on RyR2, whose sensitivity can be modulated through ROS-mediated posttranslational modifications^{9,10}. Such a positive feedback signaling transduction mechanism might allow a relatively low abundant but well sarcolemma-distributed Piezo1 to optimally respond to the drastic and repeated mechanical stress generated by the beating heart. Indeed, either deletion or overexpression of Piezo1 resulted in dysregulated Ca^{2+} and ROS signaling and heart dysfunction (revised Figs. 6, 7). Piezo1 might also contribute to maintain the SR Ca^{2+} store. Thus, we propose that the Piezo1-mediated mechano-chemo transduction process might safeguard a homeostatic functional state of the heart. Disrupting such a homeostatic role might cause abnormal contractile function and arrhythmias. For instance, breaking the positive feedback loop upon Piezo1 deletion can lead to a combination of decreased SR Ca^{2+} content and unsensitized SR Ca^{2+} release, which consequently results in compromised heart pump function and the development of cardiomyopathy. On the other hand, enforcing the positive feedback loop upon overexpression of Piezo1 might initially cause increased Ca^{2+} influx, sensitized ROS activities and enhanced spontaneous SR Ca^{2+} leakage, resulting in decreased SR Ca^{2+} store and occurrence of arrhythmogenic Ca^{2+} waves. Such a molecular mechanism might account for the severe heart failure and arrhythmia of the Piezo1-TG mice.

We have included the above discussion in the second paragraph of the Discussion section of the revised manuscript.

9. The authors showed the upregulation of Piezo1 protein in doxorubicin-induced heart failure mice hearts and human heart sample. As you know, Piezo1 protein is highly expressed in the vessel and blood. Upregulation of Piezo1 is possibly the result from the remodeling of coronary artery in pathology. The authors should compare Piezo1 mRNA normalized by internal control (ex, GAPDH) in neonatal myocytes with or without PE-treatment.

Response: We agree with the reviewer about the potential indirect effect of Piezo1 expressed in blood vessels and red blood cells on doxorubicin-induced heart failure. To test whether the upregulation of Piezo1 expression in diseased hearts might be due to an autonomic response of cardiomyocytes, we subjected cardiomyocytes derived from the littermate control and the Piezo1-KO neonatal hearts to the hypertrophic agonist phenylephrine (PE) for modeling cellular hypertrophy¹¹. PE-treated control myocytes showed increased expression of ANP (revised Fig. 8c), verifying the hypertrophic effect of PE.

Consistent with the observed upregulation of Piezo1 in diseased heart tissues (revised Fig. 8a, b), the mRNA expression of Piezo1 was also significantly increased in PE-treated control myocytes compared to non-treated control cells (revised Fig. 8c), suggesting an autonomic upregulation of Piezo1 in cardiomyocytes. Intriguingly, while the KO myocytes without PE treatment showed higher expression of ANP and β -MHC than control myocytes without PE treatment, PE-treated KO cells unexpectedly had reduced expression of ANP and β -MHC (revised Fig. 8c). These data indicate that deletion of Piezo1 appears to reverse PE-induced hypertrophy of cardiomyocytes.

10. This report suggested deletion of Piezo1 might prevent doxorubicin (DOX)-induced heart failure phenotype in the cKO mice. Since the loss of contractile function happened upon Piezo1-deletion, it is very difficult to evaluate its response to DOX. The overall phenotype was a result of defects in contractility combined with much slower (weeks) and less severe defects induced by DOX. Therefore, the conclusion about the role of Piezo1 protein in DOX-induced heart failure was not a valid one. This part of the study was not well designed and could not offer a clear insight to the Piezo1 function in DOX-induced heart failure. In this study, this is a distraction and should be removed.

Response: Given that both deletion and overexpression of Piezo1 can lead to heart dysfunction in mice (revised Fig. 6 and Fig. 7), we think it is important to examine the pathological contribution of Piezo1 to cardiomyopathy. Given that Piezo1 is upregulated in both PE-induced cellular and doxorubicin-induced mouse models of cardiomyopathy and in human patients with cardiomyopathy, we think it is reasonable to ask whether removing Piezo1 might prevent or deteriorate the progress of cardiomyopathy using the DOX model. The finding that deletion of Piezo1 might prevent or even tend to reverse doxorubicin (DOX)-induced cardiomyopathy might provide insight not only into the pathogenic importance of Piezo1 but also therapeutic treatment of human cardiomyopathy. Thus, we hope the reviewer would agree with us to keep the data in the revised manuscript.

Reviewer #2 (Remarks to the Author):

This study investigates a role for the mechanosensitive cation channel Piezo1 mechano-chemo transduction in normal and diseased hearts. The authors utilized several mouse models, including the previously established Piezo1-tdTomato knock-in model, wild-type C57/BL6 mice, and several new mouse models. New models include a Piezo-Flag knock-in reporter mouse, a cardiac-specific Piezo1 knockout mouse, and a cardiac-specific Piezo1 overexpressing mouse. Protein expression and colocalization were assessed using immunocytochemistry, immunoprecipitation, western blotting, and PCR. Heart structure, function, and remodeling were determined via histology, measurement of body weight and heart weight, echocardiography, and electrocardiography. Calcium signaling was assessed via Fura-2 single cell calcium imaging, and ROS signaling was assessed via 2',7'-dichlorofluorescein diacetate assays. The authors report that Piezo1 is expressed in the sarcolemma of adult cardiomyocytes, and that Piezo1 plays a role in mediating both stretch-induced and homeostatic calcium and ROS signaling. Additionally, they demonstrated that cardiac-specific knockout of Piezo1 impairs heart function and that cardiac-specific overexpression of Piezo1 may induce severe heart failure and arrhythmias. Finally, they report that autonomic upregulation of Piezo1 contributes to the development of cardiomyopathy. A clear strength of this paper is the use of novel mouse models. The Piezo-Flag knock-in reporter mouse, cardiac-specific Piezo1 transgenic mouse, and cardiac-specific Piezo1 knockout mouse are all excellent tools for exploring the questions asked in this paper, as well as for asking future questions. Importantly, a range of techniques are used to assess cardiac function, including echocardiography, ECG, heart weight, and fibrosis analysis, was also a strength. They used appropriate controls, and overall, their methods seemed to be appropriate, especially their assessments of calcium signaling and their induction of cell stretching. That said, there are some concerns with the approach and data interpretation that need to be addressed.

Response: We thank the reviewer for the thorough review and constructive comments for helping to improve the study.

Specific comments:

In presenting their arrhythmia data from the transgenic mice, the authors only present representative ECG traces from 4-week-old, 8-week-old, and 18-week-old mice (Fig. 5h). It would be preferable if they quantified the occurrence of arrhythmias in some way (even just incidence of VT, PVCs, or alternans), because with just the representative traces, it is difficult to appreciate the nature or extent of arrhythmia in overexpressing mouse.

Response: Following the reviewer's suggestion, we have analyzed the incidence of ventricular tachycardia in revised Fig. 7h. Remarkably, while none of the 12 littermate control mice showed arrhythmia, all the 12 TG mice examined at 4-week, 8-week or 18-week old showed ventricular tachycardia (Fig. 7h), in line with the frequently observed arrhythmogenic Ca^{2+} waves in the TG cardiomyocytes (revised Fig. 4l-n). We have clearly stated this in the Result section of "Cardiac-specific overexpression of Piezo1 induces heart failure and arrhythmias" in the revised manuscript.

There is some inconsistency in reporting of the data. For confirming protein expression in their different mouse models, they show mRNA levels and western blots for certain mice and immunostaining for others (Fig. 1a-b, e-g). To be rigorous, they should provide both sets of data (mRNA levels and western blots as well as immunostaining) for each mouse model in which they're examining expression. Additionally, DCF fluorescence data are reported for the control cardiomyocytes before and after Yoda1 treatment in the absence of 1.8 mM calcium, but they do not report this data for KO cardiomyocytes (Fig 3a-f). For some calcium signaling experiments, they report number of cells examined, but for others they report number of animals and number of cover slips. For consistency, they should report the number of cells in addition to the number of animals and cover slips, where applicable (Fig. 1h-j, Fig. 2c, e-f, h-l, l-m).

Response: Since there is currently no commercially available Piezo1 antibody suitable for detecting endogenously expressed Piezo1, it is unfeasible for us to perform immunostaining detection from control and KO mice. Therefore, we detected the Piezo1 expression in control and KO cardiomyocytes using western blotting and RT-PCR (revised Fig. 2a, b), and clearly showed that both the mRNA and protein of Piezo1 was significantly reduced in the KO cardiomyocytes.

Since Piezo1-KO cardiomyocytes did not respond to Yoda1 stimulation in the presence of 1.8 mM Ca^{2+} as shown in the revised Fig. 5e, we hope the reviewer would agree with us that it is not needed to test the effect of Yoda1 on KO cells in the absence of 1.8 mM Ca^{2+} .

Following the reviewer's suggestion, we have modified the revised Fig.2d, g, k, l; Fig. 4c, f by showing the number of cells in the revised version of the manuscript, and specified in the figure legend the number of animals and coverslips.

The authors also are not clear what sex of mice they utilized, and it is important to utilize both male and female mice to avoid confounding their data by sex effects. Also, what are background of mice used in doxorubicin study?

Response: All animal experiments were done with male mice. The Piezo1-Flag-KI mice, Piezo1-TG and Piezo1-KO mice were in C57/BL6 background. Doxorubicin-induced model were conducted in Piezo1-KO male mice and control littermates in C57/BL6 background. These information have now been clearly described in method of the revised manuscript.

Zoomed in images don't seem to align with yellow boxed regions in Figure 1a.
Response: We thank the reviewer for carefully pointing this out. We have aligned the zoomed pictures with yellow boxed regions in the revised Fig.1a in the revised version of the manuscript.

Hearts of KO appear larger by histology with increased chamber dimensions by echo but HW actually decreased in Piezo1-KO – what is the explanation?

Response: As shown in the revised Fig. 6a, the heart weight (HW) was not different between WT and KO despite that the KO heart appeared slightly larger by histology.

The authors come to the conclusion that “the mechanosensitive Piezo1 channel serves as the long-sought-after cardiac mechanotransducer that directly converts mechanical stretch of cardiomyocytes into Ca²⁺ and ROS signaling.” This seems like a bit of a lofty statement – although they demonstrate that it plays a role in translating mechanical stretch into calcium and ROS signaling, they are going a bit far to say that it is the “long-sought-after cardiac mechanotransducer.”

Response: We thank the reviewer for the critical comment. We have changed the “long-sought-after cardiac mechanotransducer” to “a key cardiac mechanotransducer” in both the Abstract and the Discussion sections.

Reviewer #3 (Remarks to the Author):

The authors report a novel role for stretch-activated ion channel Piezo1 in the slow force response (SFR) of Ca²⁺ and ROS regulation in cardiomyocytes as an in vitro equivalent to the Anrep effect. They also demonstrate cardiomyopathies associated with KO and overexpression of Piezo1 in mice.

This is a comprehensive study that represents a novel and important contribution that will be of interest to the field. In particular, the demonstration of a function effect of Piezo1 towards the Anrep effects and a potential involvement in cardiac pathologies are novel. However, some claims are not fully supported by the data and methodological and statistical clarification is needed. In particular:

Response: We thank the reviewer for the thorough review and constructive comments for helping to improve the study.

1. The authors do not account for previously-demonstrated mechanisms in the SFR, such as the crosstalk between Na/H and Na/Ca exchange (such as Luers et al. (2005) *Cardiovasc Res* (10.1016/j.cardiores.2005.07.001); and the additional citing literature that follows). The current work should be discussed in this context.

Response: We thank the reviewer for the constructive comment. We have discussed the previously-demonstrated mechanisms accounting for the SFR as the following in the revised manuscript.

“Previous studies have indicated the contribution of other Ca²⁺ signaling pathways to the development of the slow force response (SFR) of the Anrep effect⁸. For instance, on the basis of that pharmacological blocking of the Na⁺/H⁺ exchange (NHE1) reduced the SFR by ~50%, it has been hypothesized that stretch-dependent modulation of NHE1 increases intracellular Na⁺, which in turn activates the reverse mode of the Na⁺/Ca²⁺ exchange (NCX) to increase intracellular Ca²⁺, leading to the development of SFR¹². However, it remains unclear whether NHE1 directly respond to mechanical stimulation or indirectly affects the bona fide cardiac mechanotransducer. Piezo1 has been shown to be inhibited by low pH¹³. One conceivable scenario is that inhibition of NHE1 results in decreased pH, which in turn inhibits stretch-induced and Piezo1-mediated Ca²⁺ influx. Furthermore, as a cationic channel, Piezo1 conducts Na⁺ as well, raising the possibility that Piezo1 might help to raise intracellular Na⁺ and consequently activate the reverse mode of NCX to increase intracellular Ca²⁺. Alternatively, Piezo1 and NHE1-NCX might function in parallel mechanotransduction pathways. Indeed, inhibition of NHE1 and NCX only suppressed ~50% of the SFR. Future studies might dissect out the relationship between Piezo1 and other molecules involved in cardiac mechanotransduction.”

2. The authors state “Here we have established that the mechanosensitive

Piezo1 channel serves as the long-sought-after cardiac mechanotransducer that directly converts mechanical stretch of cardiomyocytes into Ca²⁺ and ROS signaling”, but do not adequately demonstrate a contribution to the Frank-Starling (or rapid) stretch response and do not acknowledge the other demonstrated mechanisms of Ca stretch response. Claims should be tempered and qualified.

Response: We thank the reviewer for the constructive comment. We have changed “the long-sought-after cardiac mechanotransducer” to “a key cardiac mechanotransducer” in both the Abstract and the Discussion sections. As answered in the above comment 1, we have discussed previously demonstrated mechanisms of Ca²⁺ stretch response.

3. “The observation that a stretched diastolic left ventricle of the KO hearts ($81.8 \pm 8.7 \mu\text{l}$) relative to that of the control hearts ($61.2 \pm 3.4 \mu\text{l}$) led to a comparable stroke volume suggests a compromised Frank-Starling response in the KO heart. The KO hearts had a concomitant decreased ejection fraction and fractional shortening”. This is a description of DCM phenotype. Without testing each condition’s response to an acute preload, a compromised F-S effect is not a logically-sound argument. Even then, the changes to anatomy based on the DCM-like phenotype provide a confounding factor to these measurements.

Response: We agree with the reviewer’s comment. Accordingly, we have changed the description to “The observation that a stretched diastolic left ventricle of the KO hearts ($81.8 \pm 8.7 \mu\text{l}$) relative to that of the control hearts ($61.2 \pm 3.4 \mu\text{l}$) led to a comparable stroke volume suggests a compromised stroke function in the KO heart. The KO hearts had a concomitant decreased ejection fraction and fractional shortening” in the revised manuscript.

4. The stretch experiments were performed with 8% longitudinal strain, which is well below the 15-20% that would represent a healthy beat. This may underscore the predominance of Piezo1 in the SFR demonstrated in this paper and should be discussed.

Response: Following the reviewer’s suggestion, we have applied a 15% of stretch that is within the range of a diastolic stretch normally occurred during a healthy heart beat. Compared to the 8% stretch-induced response, 15% stretch induced a relatively more robust increase in Ca²⁺ sparks, which could still largely recover upon relaxation (new Fig. 3d, e), indicating no major damage to the cells. Importantly, such responses were not observed in the Piezo1-KO cardiomyocytes (new Fig. 3d, e). The results have been included in new Fig. 3d, e and the description have been included in the Result section of “Piezo1 mediates stretch-induced Ca²⁺ sparks in cardiomyocytes” in the revised manuscript.

5. The composition of buffers used in experiments must be defined, given the relationships between ionic environment and the function of ionic channels.

Additionally, confirm “50 mM Ca²⁺ Tyrode’s solution containing 0.9 mg/mL type II collagenase”. Was this supposed to be 50 mg/L? (approx. 33% of physiological Ca²⁺)? Please provide composition of Tyrode’s solution used as most dissociations use Ca-free, Mg-substituted media with collagenase; if Ca was indeed at 50 mM, please provide solubilization steps and measures to balance osmolarity.

Response: We thank the reviewer for the careful review and apologize for the mistake. We have corrected the calcium buffer concentration to “50 μM” in the Methods in the revised version of the manuscript.

6. Piezo1 knockout had significantly lower baseline levels of Ca²⁺ sparks (Fig. 2A,B), confounding the interpretation of the lack of stretch-response and lack of stretch-induced ROS production. Does stretch induce Ca²⁺ sparks in TG cells subjected to stretch? Are Ca²⁺ sparks inducible in the KO cells by non-mechanical stimuli?

Response: We have attempted to detect stretch-induced Ca²⁺ sparks in Piezo1-TG cardiomyocytes. However, we found that the TG cell were highly vulnerable to the process of attaching the electrode to the cell for stretch stimulation, which might indicate highly sensitized responsiveness of the TG cells to mechanical manipulation. Due to this technique difficulty, we could not complete the experiment to assay stretch-induced Ca²⁺ sparks in TG cells.

To exclude the possibility that the lack of stretch-induced Ca²⁺ spark in the KO cells might be due to their decreased SR Ca²⁺ content, we employed 1 mM caffeine to trigger Ca²⁺ sparks (new Fig. 3f, g). Despite relatively lower Ca²⁺ spark events in the KO cardiomyocytes due to their reduced SR Ca²⁺ content, the caffeine-induced fold change of Ca²⁺ sparks were similar between the control and KO cells (new Fig. 3g). These data suggest that the Piezo1-KO cardiomyocytes specifically lost stretch-induced but retained caffeine-induced Ca²⁺ sparks. Taken together, these data demonstrate that Piezo1 mediates stretch-induced Ca²⁺ sparks in cardiomyocytes. We have included these new data in Fig. 3f, g and described the results in the Result section of “Piezo1 mediates stretch-induced Ca²⁺ sparks in cardiomyocytes”.

7. The proposed pathway in 3i is not fully supported by the data presented. In particular, do Yoda1 or stretch activate Nox2 or ROS and to similar degrees?

Response: The measurement of ROS production induced by Yoda1 or stretch was conducted under different experimental conditions. For instance, Yoda1-induced production of ROS was analyzed by comparing the fold change of whole-cell DCF fluorescence change before and after Yoda1 application. However, line-scan imaging of local DCF fluorescence was carried out to measure stretch-induced production of ROS. Thus, it is unfeasible to directly compare the degree of Yoda1 or stretch-induced ROS signals. However, since both Yoda1 and stretch-induced change of DCF fluorescence were nearly

completed abolished in the Piezo1-KO cardiomyocytes, we hope the reviewer would agree with us that the proposed pathway in the revised Fig. 5i is supported by our data.

8. The caption in Fig 1J implies that multiple coverslips were used from a single mouse and treated as independent samples in the statistical analysis. Please clarify and justify.

Response: Following both reviewer 2 and 3's suggestion, we have changed the figures by showing individual cells and stated the number of coverslips and mice used in the figure legends.

9. The normalization employed in Fig 3j and 3k yields no variance in the "before" samples and thus inappropriate t-tests. Please clarify the normalization procedure in the methods, or reanalyze data for j and k as was done previously in Fig 3b,c,d,f,g.

Response: Following the reviewer's suggestion, we have reanalyzed the data and shown in the revised Fig. 5j, k.

10. In Supplementary Fig 4: These analyses represent multiple comparisons. They should be presented in a single panel, and should be analyzed by 2-way ANOVA followed by a post-hoc test of choice. This would also carry a benefit in assessing main effects and the potential for interaction.

Response: Following the reviewer's suggestion, we have reanalyzed the data and shown in the revised Supplementary Fig. 4.

11. Patient characteristics for human heart tissue are missing. What types of CM are represented in the disease group?

Response: The human heart tissues were derived from patients with hypertrophic obstructive cardiomyopathy. The human sample information has been included in the Supplementary Excel Sheet in the revised manuscript.

Reviewer #4 (Remarks to the Author):

The beating heart possesses intrinsic ability to adapt cardiac output to changes in mechanical load. The century-old Frank-Starling law and Anrep effect have documented that stretching heart during diastolic filling increases its contractile force. However, the molecular mechanotransduction mechanism and its impact on cardiac health and disease remain elusive. Here we show that the mechanically activated Piezo1 channel converts mechanical stretch of cardiomyocytes into Ca²⁺ and reactive oxygen species (ROS) signaling, which critically determines the mechanical activity of the heart. Either cardiac-specific knockout or overexpression of Piezo1 in mice results in defective Ca²⁺ and ROS signaling and the development of cardiomyopathy, demonstrating a homeostatic role of Piezo1. Piezo1 is pathologically upregulated in both mouse and human diseased hearts, and its deletion renders mice with dilated cardiomyopathy resistant to deterioration of cardiac function. Thus, Piezo1 serves as the long-sought-after cardiac mechanotransducer for initiating mechano-chemo transduction and consequently maintaining normal heart function, and might represent a novel therapeutic target for treating human heart diseases. Also this work is interesting, a number of concerns remain.

Response: We thank the reviewer for the thorough and positive review of our work.

1. The degree (extent) of stretch makes any difference? A couple of more points may be worthy trying.

Response: Following both reviewer 3 and 4's suggestion, we have applied a 15% of stretch that is within the range of a diastolic stretch normally occurred during a healthy heart beat. Compared to the 8% stretch-induced response, 15% stretch induced a relatively more robust increase in Ca²⁺ sparks, which could still largely recover upon relaxation (new Fig. 3d, e), indicating no major damage to the cells. Importantly, such responses were not observed in the Piezo1-KO cardiomyocytes (new Fig. 3d, e). These new results have been included in new Fig. 3d, e and the description have been included in the Result section of "Piezo1 mediates stretch-induced Ca²⁺ sparks in cardiomyocytes" in the revised manuscript.

2. What is the rationale of using 18 weeks? Any data for survival for the overexpression mice?

Response: We have initially characterized 8-week-old mice, but found the Piezo1-KO mice had no obvious defects in their heart morphology and pump function (Supplementary Fig. 3a-e, Supplementary Table 1a and Supplementary Movie1). We have therefore decided to examine 18-week-old mice.

We did not systematically obtain the data for survival for the Piezo1-TG mice. Nevertheless, we did notice that the TG mice died occasionally under normal feeding conditions, and became faint or died upon grasping or changing cages.

3. What may be the precise mechanism for Piezo1-induced calcium regulation?

Response: The observation that Piezo1 controls both stretch-induced and homeostatic Ca^{2+} and ROS signaling might suggest a Piezo1-mediated positive feedback loop between these two signaling molecules critical for cardiac function. Piezo1-dependent Ca^{2+} influx regulates the production and homeostasis of ROS via the Rac1-NOX2 signaling pathway, which might in turn modulate Ca^{2+} release via acting on RyR2, whose sensitivity can be modulated through ROS-mediated posttranslational modifications^{9,10}. Such a positive feedback signaling transduction mechanism might allow a relatively low abundant but well sarcolemma-distributed Piezo1 to optimally respond to the drastic and repeated mechanical stress generated by the beating heart. Indeed, either deletion or overexpression of Piezo1 resulted in dysregulated Ca^{2+} and ROS signaling and heart dysfunction (revised Figs. 6, 7). Piezo1-mediated Ca^{2+} influx might also contribute to maintain the SR Ca^{2+} store. Thus, we propose that the Piezo1-mediated mechano-chemo transduction process might safeguard a homeostatic functional state of the heart. Disrupting such a homeostatic role might cause abnormal contractile function and arrhythmias. For instance, breaking the positive feedback loop upon Piezo1 deletion leads to a decreased Ca^{2+} influx and ROS production, resulting in a combination of reduced SR Ca^{2+} content (revised Fig. 2f, g) and unsensitized SR Ca^{2+} release (revised Fig. 4a-c), which consequently causes compromised heart pump function and the development of cardiomyopathy of the Piezo1-KO mice (revised Fig. 6). On the other hand, enforcing the positive feedback loop upon overexpression of Piezo1 might initially cause increased Ca^{2+} influx (revised Fig. 2j, k), sensitized ROS activities (revised Fig. 5m) and enhanced spontaneous SR Ca^{2+} leakage (revised Fig. 4d, f), resulting in decreased SR Ca^{2+} store (revised Fig. 2i) and occurrence of arrhythmogenic Ca^{2+} waves (revised Fig. 4g-i). Such a molecular mechanism might account for the severe heart failure and arrhythmia of the Piezo1-TG mice (revised Fig. 7). We have included the above discussion in the Discussion section of the revised manuscript.

4. Any effects on mitochondrial calcium – any ultrastructural changes?

Response: Since we found that Piezo1 is localized in the sarcolemma and mediates Ca^{2+} influx, we believe that Piezo1 might not directly affect mitochondrial Ca^{2+} . Therefore, we hope the reviewer would agree with us that the study of the effect of Piezo1 on mitochondrial Ca^{2+} and potential ultrastructural changes secondary to Piezo1 knockout or overexpression might be beyond the major focus of the present study.

References

- 1 Coste, B. *et al.* Piezo1 and Piezo2 are essential components of distinct mechanically activated cation channels. *Science* **330**, 55-60, doi:10.1126/science.1193270 (2010).
- 2 Liang, J. *et al.* Stretch-activated channel Piezo1 is up-regulated in failure heart and cardiomyocyte stimulated by AngII. *American journal of translational research* **9**, 2945-2955 (2017).
- 3 Blythe, N. M. *et al.* Mechanically activated Piezo1 channels of cardiac fibroblasts stimulate p38 mitogen-activated protein kinase activity and interleukin-6 secretion. *The Journal of biological chemistry* **294**, 17395-17408, doi:10.1074/jbc.RA119.009167 (2019).
- 4 Syeda, R. *et al.* Chemical activation of the mechanotransduction channel Piezo1. *eLife* **4**, doi:10.7554/eLife.07369 (2015).
- 5 Wang, Y. *et al.* A lever-like transduction pathway for long-distance chemical- and mechano-gating of the mechanosensitive Piezo1 channel. *Nature communications* **9**, 1300, doi:10.1038/s41467-018-03570-9 (2018).
- 6 Prosser, B. L., Ward, C. W. & Lederer, W. J. X-ROS signaling: rapid mechano-chemo transduction in heart. *Science* **333**, 1440-1445, doi:10.1126/science.1202768 (2011).
- 7 Iribe, G. *et al.* Axial stretch of rat single ventricular cardiomyocytes causes an acute and transient increase in Ca²⁺ spark rate. *Circulation research* **104**, 787-795, doi:10.1161/CIRCRESAHA.108.193334 (2009).
- 8 Khairallah, R. J. *et al.* Microtubules underlie dysfunction in duchenne muscular dystrophy. *Sci Signal* **5**, ra56, doi:10.1126/scisignal.2002829 (2012).
- 9 Terentyev, D. *et al.* Redox modification of ryanodine receptors contributes to sarcoplasmic reticulum Ca²⁺ leak in chronic heart failure. *Circulation research* **103**, 1466-1472, doi:10.1161/CIRCRESAHA.108.184457 (2008).
- 10 Ward, C. W., Prosser, B. L. & Lederer, W. J. Mechanical stretch-induced activation of ROS/RNS signaling in striated muscle. *Antioxidants & redox signaling* **20**, 929-936, doi:10.1089/ars.2013.5517 (2014).
- 11 Glennon, P. E. *et al.* Depletion of mitogen-activated protein kinase using an antisense oligodeoxynucleotide approach downregulates the phenylephrine-induced hypertrophic response in rat cardiac myocytes. *Circulation research* **78**, 954-961, doi:10.1161/01.res.78.6.954 (1996).
- 12 Luers, C. *et al.* Stretch-dependent modulation of [Na⁺]_i, [Ca²⁺]_i, and pHi in rabbit myocardium--a mechanism for the slow force response. *Cardiovascular research* **68**, 454-463, doi:10.1016/j.cardiores.2005.07.001 (2005).
- 13 Bae, C., Sachs, F. & Gottlieb, P. A. Protonation of the human PIEZO1 ion channel stabilizes inactivation. *The Journal of biological chemistry* **290**, 5167-5173, doi:10.1074/jbc.M114.604033 (2015).

REVIEWERS' COMMENTS

Reviewer #1 (Remarks to the Author):

The revised version of the manuscript has been significantly improved. I have no additional concerns.

Reviewer #2 (Remarks to the Author):

The authors have been responsive in their revised manuscript - there are no outstanding concerns.

Reviewer #3 (Remarks to the Author):

The authors' substantive revisions adequately address my comments.

Reviewer #4 (Remarks to the Author):

The authors have addressed a number of the concerns raised in the initial round. However, several aspects remain sub-standard.

1. The authors' reply to why KO mice offer beneficial effect against doxorubicin while eliciting detrimental effect itself is not satisfactory.

2. The initial question of age (no phenotype at 8 weeks and cardiomyopathy at 18 weeks) remains unanswered.

Reviewer #4 (Remarks to the Author):

The authors have addressed a number of the concerns raised in the initial round. However, several aspects remain sub-standard.

Response: We thank the reviewer for the further comments of the manuscript.

1. The authors' reply to why KO mice offer beneficial effect against doxorubicin while eliciting detrimental effect itself is not satisfactory.

Response: In light of the confusion of the data brought to the reviewer, we have removed all the relevant data and corresponding interpretations from the original Figure 8 d-g, Supplementary Table 1f (KO data) and Supplementary Figure 4.

2. The initial question of age (no phenotype at 8 weeks and cardiomyopathy at 18 weeks) remains unanswered.

Response: We have clearly stated in the Discussion section of the revised manuscript that "Despite that the exact reason for the age-dependent phenotype remains unclear, we currently speculate that such a hypothesis might explain why only Piezo1-KO mice at 18-week old but not at 8-week old developed cardiomyopathy (Fig. 6)."